



# Experimental demonstration of regenerative wind farming using a high-density layout of VAWTs

David Bensason[1], Jayant Mulay[1], Andrea Sciacchitano[1], and Carlos Ferreira[1]

[1]Flow Physics and Technology, Faculty of Aerospace Engineering, Delft University of Technology, Netherlands

**Correspondence:** David Bensason (d.y.bensason@tudelft.nl)

**Abstract.** The present study extends the idea of the VAWT "vortex generator mode" for wake recovery on a wind farm scale, working towards the concept of "regenerative wind farming", where upstream turbines entrain vertical momentum for those downstream. An experimental wind tunnel demonstration of the "regenerative wind farming" concept for an array of nine H-type VAWTs arranged in a 3x3 grid layout is performed. Volumetric particle tracking velocimetry measures the wake within the simulated wind farm while using two "vortex generator modes" achieved through fixed blade pitch. The results demonstrate the strong dependence of the wake topology of a VAWT on the streamwise vorticity system, which can be effectively modified by pitching the blades and subsequently changing the load distribution of the different quadrants of a VAWT. An increase in momentum entrainment in the wake is observed for both "vortex generator modes" of operation, highlighting the potential towards the goal of "regenerative wind farming." The derived available power within the farm increases by factors of 6.4 and 2.1 for the pitch-in and -out cases compared to the baseline case, respectively, considering potential rotors directly in line with those upwind.

## 1 Introduction

Offshore wind energy research and deployment have gained popularity in recent years, given the favorable wind resources realized by offshore turbines (Möller et al. (2012)). Furthermore, the trend for both onshore and offshore wind plant projects has been to follow a cluster arrangement, given the benefits of having closely spaced rotors from economic, operational, and maintenance perspectives (Sørensen and Larsen (2021); Shields et al. (2021)). This has also become a requirement for offshore applications, given the limited space for installations that include lucrative wind resources, the possibility for bottom fixed foundations, and limited social and political restrictions. Although the clustering of turbines has benefits on the operational side, it has been established through field, experimental, and numerical studies that densely packed turbines often experience a penalty in performance due to wake losses, leading to the underperformance of the wind plant as a whole (Barthelmie et al. (2010)). The severity of these losses is strongly dependent on the layout of the farm itself as well as the atmospheric conditions at the respective location. Field studies of offshore farms have demonstrated losses ranging from 10% to 23% depending on the turbine spacing (ranging from 4D to 12D) and wind resource (Barthelmie et al. (2007)).

The wake recovery process for an isolated HAWT has been studied extensively, both experimentally and numerically (see reviews by Sørensen (2011); Porté-Agel et al. (2020)). In the near wake, where coherent tip and root vortices are identifiable,





the entrainment of energy is ascribed to a vertical advection of momentum, accompanied by vortex leapfrogging. However, once these coherent structures break down, the process is mainly governed by turbulent kinetic energy entrainment, leading to phenomena such as wake meandering (Porté-Agel et al. (2020); Lignarolo et al. (2015)). Hence, within a wind farm, the main driver of the replenishment of momentum in the wake is governed by the rate of turbulent entrainment of kinetic energy

(Stevens and Meneveau (2017)).

The greatest potential for replenishing the kinetic energy in a farm is to entrain high momentum flow from above the rotor layer, as the turbines are often situated close to the sea surface. Given the importance of wake losses on the performance and overall efficacy of a wind plant, several strategies have been developed to mitigate them. Wake steering is a heavily researched and applied method for increasing the net performance of a wind plant through the intentional misalignment of selected turbines

with the incoming flow direction, often realized through a yaw misalignment of the rotor face with the incoming wind (Fleming et al. (2017)). The "steering" turbine will forfeit some performance efficiency to deflect its wake away from downstream rotors, leading to reported performance increases of the overall wind plant (Howland et al. (2019); Fleming et al. (2020)). The potential gains through wake steering depend on factors such as wind resource (wind direction, speed, turbulence, etc.), turbine layout, and intrinsic baseline wake loss (Simley et al. (2021)). Like yaw control, utilizing the tilt angle to deflect the wake vertically is

an extensively studied method for increased wake recovery rate (Kasper and Stevens (2024); Cossu (2021)).

The underlying driver of enhanced entrainment of kinetic energy in the wake through yaw and tilt misalignment has been attributed to a counter-rotating vortex pair (CVP) generated due to the introduction of a lateral force on the wake (Bastankhah and Porté-Agel (2016)), which enhance the rate of advective flow re-energizing in the wake. This phenomenon has been further demonstrated experimentally by Bossuyt et al. (2021) for both a yawed and tilted rotor configuration, highlighting the

mechanism of wake recovery using the Reynolds averaged Navier-Stokes equation (RANS) terms. In the case of wake steering using yaw misalignment, the influx of high-momentum flow is realized from the sides of the rotor, with the wake being deflected to the side. Tilting the rotor is another wake control strategy that has been investigated and relies on the same underlying physics as the yawed case, with an included vertical force component and CVP, which entrain high momentum flow from above and below the rotor while steering the wake upward or downward. Recent studies (Kasper and Stevens (2024)) have demonstrated

the potential of this wake steering technique for large-scale wind farms. Despite the promise of these wake steering strategies, studies have commented on the impact of yaw control on a farm level on the fatigue loading of the turbines. The work by Shaler et al. (2022) concluded that although yawing turbines in a farm that operates at and below rated conditions may not drive the ultimate loads, the fatigue loads will increase overall and become more uniform throughout the farm. Nevertheless, the relative impacts on blade-root and shaft-bending modes are heavily dependent on the wind conditions and frequency of yaw control.

In contrast to the long wake lengths of HAWTs, vertical-axis wind turbines (VAWTs) have demonstrated an intrinsic acceleration in kinetic energy replenishment in the wake ranging up to 6D of the rotor (Wei et al. (2021)). Despite their lower reported power performances, research into closely spaced, co, and counter-rotating configurations has demonstrated an order of magnitude increase in performance due to the favorable fluid interactions between the turbines (Vergaerde et al. (2020); De Tavernier et al. (2018)). These findings have stimulated an increased effort towards studying the three-dimensional VAWT wakes as they

have the potential for increasing the power density of wind farms compared to the conventional HAWT arrangements (Dabiri





(2011)). The wake characteristics and dynamics of VAWTs have been extensively researched through experimental and numerical investigations. Similar to HAWTs, the wake recovery process of VAWTs is characterized by the turbulent entrainment in kinetic energy from above and below the wake. However, the enhanced rate of this recovery with respect to VAWTs is mainly attributed to the presence of dominant streamwise vortical structures shed by the blade tips throughout the periodic cycle of the

rotor (Huang et al. (2023b)), which entrain momentum through an advective process. Numerical simulations by Boudreau and Dumas (2017) demonstrated the persistence of these advective contributions towards the wake replenishment in the wake for a cross-flow turbine compared to an axial turbine.

A recent study by Huang et al. (2023b) experimentally demonstrated the relationship between the VAWT load distribution and trailing vorticity structures. By simplifying the load distribution to an idealized actuator cylinder surface (Madsen et al.

(2014)), the deformation and deflection of the VAWT wake topology were linked to the distribution of load discretized between the different quadrants of the rotor cycle. By redistributing the load amongst the upwind, downwind, windward, and leeward quadrants, the strength of the trailing vortical structures shed in the respective quadrants can be increased or decreased, leading to modification of the wake topology. This concept was further developed and confirmed via numerical simulations (De Tavernier et al. (2020); De Tavernier and Ferreira (2019)). An effective method to modify the load cycle of a VAWT is through

blade pitch. Experimental results by LeBlanc and Ferreira (2022b) demonstrated the impact of cyclic and fixed blade pitch motions on the normal load and subsequent thrust vectoring of an H-type VAWT, confirming the augmentation and redistribution of load amongst the different quadrants of the cycle. Subsequent planar flowfield measurements at the mid-span of the rotor blade at different phases confirmed the normal load distribution with varying pitching schemes (LeBlanc and Ferreira (2022a)). The work of Huang et al. (2023b, a) measured the wake of an H-type VAWT using a stereoscopic measurement system at dis-

crete downwind plans ranging up to 10D with different fixed pitch offsets applied to the blades, namely a case where blades are pitched inward (inside the rotor swept volume) and outward. A comprehensive visualization of the wake deformation and deflection as a function of the augmented trailing vorticity was revealed, demonstrating significant gains in available power for potential downstream rotors of up to 150% at 5D. An enhanced advective momentum flux above and below the rotor was observed for the case where the load is shifted towards the upwind half of the blade cycle. In contrast, the lateral advective

kinetic energy entrainment is increased when the load is shifted towards the downward half.

Current experimental studies, which are 2D2C or 2D3C, provide limited information on the three-dimensionality of a VAWT wake topology and vortex strength due to the intrinsic limitations of planar measurements. This is further magnified by the use of blade pitching schemes, which rely mainly on the modification of these vortical structures, leading to heavy 3D wake phenomena (De Tavernier et al. (2020)). The literature on experimental 3D-resolved wakes of VAWTs in wind tunnel settings

is limited (Brownstein et al. (2019); Caridi et al. (2016); Ryan et al. (2016); van der Hoek et al. (2024)) and is not available for rotors using the aforementioned wake control strategy. Further, the experimental demonstration with a 3D resolved flowfield of the potential of blade pitch towards wake re-energization in a farm setting has not been investigated to date. The work of Brownstein et al. (2019) yielded 3D flowfield measurements for a set of two turbines closely spaced to highlight the benefits of high-density VAWT arrangements but was limited to a wake measurement domain up to 3D.





The goal of this paper is to report the 3D-resolved wake measurement results of a wind tunnel-scaled high-energy density wind farm made of VAWTs with and without wake control strategies in a standard grid arrangement. The results from this study provide a basis for ongoing and future numerical model validation. The flow fields at the inflow, within, and outflow of the farm are measured using three-dimensional Lagrangian Particle Tracking, focusing on the central column of the farm. This work offers a detailed analysis of the vortical structures in the wind farm and how the wake topology changes when these structures are enhanced. An overview of the working principle of the "vortex generator" mode of VAWTs and its application towards regenerative wind farming is provided in Section 2. The 3D wake measurement technique and wind tunnel simulated wind farm setup are described in Section 3. The discussion of the results given in Section 4 focuses on the modification of the trailing vorticity system and the subsequent impact on the wake topology. The wake recovery is quantified, and the driving mechanisms for recovery are isolated via the energy budget analysis.

## 2 Working principle of the vortex-generator mode

The working principle of an H-Type VAWT as a vortex generator has been numerically and experimentally explored by Huang et al. (2023b, a), in which the link between the wake topology and deflection to the quadrant discretized blade load distribution is evaluated.

The working principle can be best described by simplifying the phase dependent VAWT blade load as an idealized Actuator Cylinder (AC) that includes radially dependent normal loads which represent the blade integrated loads (Madsen et al. (2014)), as visualized in Figure 1. The AC simulated normal loading and wake characteristics for VAWTs have been validated using high-fidelity models and experimental results by De Tavernier et al. (2020); De Tavernier and Ferreira (2019); Martinez-Ojeda et al. (2021). The work of Huang et al. (2023b) demonstrated that a modification in the idealized load distribution of the AC could lead to enhanced wake deformation and deflection depending on the load distribution between the different quadrants. The load profile of the AC can be modified in several ways, including pitching the struts connecting the blades to the tower (Mendoza and Goude (2019); Ribeiro et al. (2024)), inclining the angle of the rotor blade (Guo and Lei (2020)), and applying blade pitch (LeBlanc and Ferreira (2022b)). The present work uses the blade pitch method for modifying the idealized AC load. As a proof of concept for using this "vortex generator" mode for a wind tunnel simulated wind farm, fixed-pitch offsets are used (constant applied throughout the entire rotor cycle), as in the work of Huang et al. (2023b).

A top-view schematic representation of the idealized actuator cylinder load for the "vortex generator" cases is shown in Figure 1, adapted from the work of Huang et al. (2023b), for a rotor rotating counter-clockwise. Three cases are shown: the baseline, where no fixed pitch is applied; the pitch-in case, where the blade is pitched towards the supporting tower; and the pitch-out, where the blade is pitched away from the rotor volume. The VAWT cycle discretized in four quadrants, namely the upwind windward UW $0° < \theta < 90°$, upwind leeward UL $90° < \theta < 180°$, downwind leeward DL $180° < \theta < 270°$, and downwind windward DW $270° < \theta < 360°$. The actuator circumference (corresponding to the upper blade-tip trajectory) is color-coded based on the direction of the trailing tip-vortex shed from each quadrant in the inertial frame of reference, with red and blue as anti-clockwise and clockwise, respectively. This cyclic variation in tip-vortex direction is intrinsic to VAWT



WIND
ENERGY
SCIENCE
DISCUSSIONS

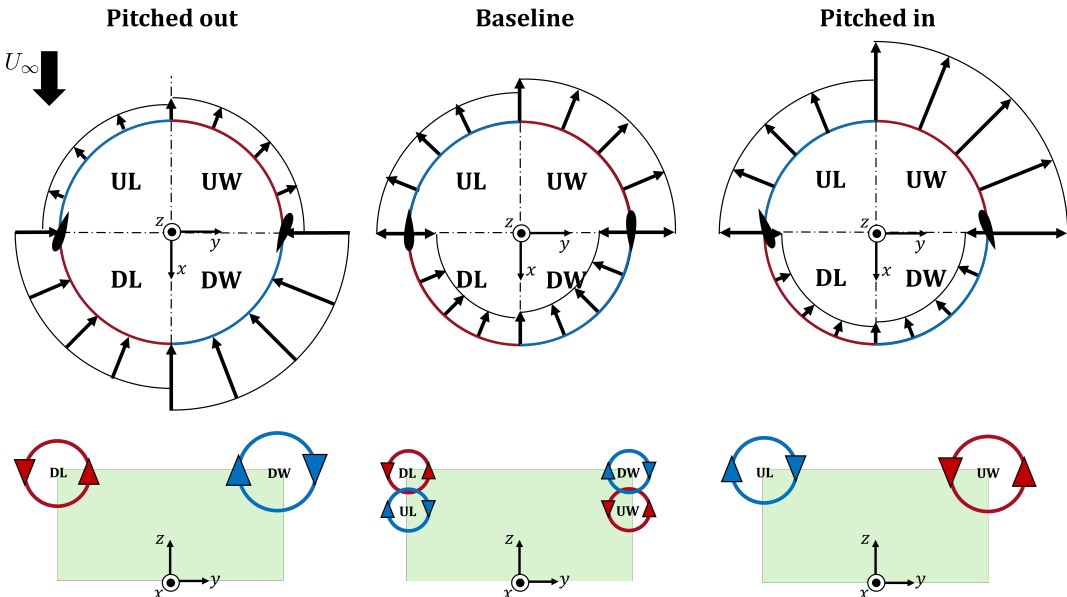

**Figure 1.** Top view schematic of the simplified AC force fields for the three "vortex generator" modes of operations adapted from Huang et al. (2023b); Bensason et al. (2024), namely the baseline case (where no pitch is applied), a pitched in case, and pitched out. Blades at a phase position of $\theta = 0°$ are shown at exaggerated sizes and pitch angles to demonstrate the pitching convention. The coordinate systems are shown at the center of the AC, and the rotation direction is anti-clockwise. The quadrants of the idealized AC are labeled as upwind windward UW $0° < \theta < 90°$, upwind leeward UL $90° < \theta < 180°$, downwind leeward DL $180° < \theta < 270°$, and downwind windward DW $270° < \theta < 360°$. The top edges of the rotor cylinder are color-coded based on the direction of rotation in the internal frame of reference of the trailing blade-tip vortices, with red and blue as counter-clockwise and clockwise, respectively. The top half of the rotor's projected frontal area (green) at a generic measurement plane downstream is shown below each mode. The projected dominant quadrant-labeled streamwise vortical structures stemming from the blade tips for each pitch case are shown via circular arrows.

aerodynamics and is a product of the variation in relative angle of attack and, subsequently, the direction of circulation of the blade-tip vortex (Tescione et al. (2014); Bensason et al. (2023)). The normal arrows represent the blade-integrated loads. Below

each mode, the projected frontal area of the top half of the rotor ($z/D > 0$) (green-shaded region) at a generic downstream plane is shown with the projected intersections of the quadrant-labeled trailing vortical structures. The directions of rotation of these structures are indicated by arrows embedded in the idealized circular vortex intersection. As the inflow is uniformly laminar and no ground effect is included in this study, the wake topology should be symmetric about the symmetry plane $z/D = 0$. Hence, the vortical structures stemming from the bottom blade tips will be equal and opposite to those generated at

the top, in accordance with Hemholz theorem.

    The AC loads shown for a VAWT are simplified to understand the trailing vorticity and subsequent wake topology described in the results that will follow. In reality, the blade load is a result of both the normal and tangential loads, of which the normal is dominant, which varies azimuthally throughout the cycle (LeBlanc and Ferreira (2022b)). Furthermore, under the realistic





operating conditions of a VAWT, where unsteady effects are present, such as dynamic stall, turbulence, and dynamic inflow,
the load profiles can change significantly. Experimentally acquired blade load profiles as a function of azimuth with variable
pitch control are presented by LeBlanc and Ferreira (2022b) for a straight-bladed VAWT and highlight this non-uniformity.
Nonetheless, these schematics are sufficient for describing the impacts of AC load modification on the wake, with each case
described as follows:

**Baseline case**:  In the case where no pitch is applied, the AC loading is intrinsically asymmetric. The upwind half of the
cycle will experience a higher load than the downwind as the latter involves the blades operating in their own wake (LeBlanc
and Ferreira (2022a); De Tavernier et al. (2020)). Furthermore, the windward passage of the blade will exhibit larger loads
than the leeward due to the higher relative wind speed as the blade moves into the wind. Despite the more favorable load of the
upwind quadrants, the trailing vorticity in the near wake of the rotor will exhibit the counter-rotating vortex pairs (CVP) from
both halves of the cycle (Huang et al. (2023b); Bensason et al. (2023); Tescione et al. (2014)). The aforementioned favorable
loading on the windward passage on the blade leads to an intrinsic asymmetry in the wake in favor of the windward side, as
shown experimentally by Tescione et al. (2014); Bensason et al. (2024).

**Pitched in case**: When the blades are pitched inward, the load is shifted towards the upwind half, as shown by the increased
integrated load magnitude along the upwind quadrants. As a result, the vortical structures generated in the upwind half are
energized and become the dominant CVP in the wake, as shown in the projected frontal area, with the less energized pairs
stemming from the downwind half charging in direction and merging with the dominant pair, as experimentally demonstrated
through phase-locked wake measurements of the near wake of a VAWT by Bensason et al. (2024). Once again, the windward
side is more energized than the leeward, leading to a stronger UW than the UL vortex, visualized by the difference in vortex
size. The impact of this load modification is twofold. Firstly, the domain CVP on the upper tips will induce a significant axial
downwash in the wake, with high momentum flow from above the rotor being injected into the volume while low momentum
wake is ejected out laterally. Secondly, given the imbalance in load between the windward and leeward halves of the rotor,
a strong lateral force will be imposed on the wake, leading to a lateral deflection of the low momentum flow towards the
windward side. The results of Huang et al. (2023b) have confirmed this mode of operation, highlighting the potential of the
enhanced advective momentum transfer between the freestream and wake towards an accelerated wake recovery.

**Pitched out case**:  Finally, when the blade is pitched out, the AC load is shifted toward the downwind half of the cycle.
Hence, the rotation directions of the vortical structures shed from the blade tips will be opposite those for the positive pitch
case. As in the aforementioned mode of operation, the dominant vortical structures in the wake will lead to a modification in
topology, with an induced upwash and downwash in the upper and lower halves, respectively, while high-speed momentum
is injected from the sides of the rotor. Meanwhile, the load imbalance between the leeward and windward sides results in a
lateral deflection of the wake center towards the leeward side. This mode of operation was also experimentally demonstrated by
Huang et al. (2023b); however, it showed to have less potential than the pitched-in mode of operation with respect to streamwise
momentum recovery, presumably due to the intrinsically lower efficiency in the downwind half of the cycle, as discussed for
the baseline case.





A schematic representation of an array of three uniformly spaced inline VAWTs is shown in Figure 2 for the three modes of operation. The wake structures at the midplane $y/D = 0$ are represented via gray-shaded areas, with the streamwise velocity profiles at different downstream locations $U_x(x)$ represented via arrows of varying size (magnitude). For this study, the inflow is uniform laminar with speed $U_\infty$ without a ground effect. Hence, the wake structure and re-energization mechanism will be symmetric about the symmetry plane in $xy$. Additionally, arrows representing advective and turbulent entrainment of kinetic energy in the wake are represented via purple and blue arrows, respectively. The sizes of these arrows reflect the relative magnitude of the contributions in the wake.

For the baseline case, in the near-wake, the wake re-energization is mainly caused by the lateral momentum advection due to the cohesive vortical structures shed in the wake, as included in Figure 1 (Boudreau and Dumas (2017)). As the structures diffuse and mix, the wake recovery process will transition towards a turbulent kinetic energy entrainment. With no strong lateral force induced by the rotor load, the wakes of different turbines will continue to superimpose on each other within the farm, growing in size due to the wake expansion.

For the negative pitch case, the advective contribution towards the wake recovery will significantly increase due to the dominant vortical structures highlighted in Figure 1. Given the direction of rotation of the downwind tip-vortex structures, high momentum flow will be injected in the wake from the lateral direction, while the low momentum wake is ejected out above and below the rotor. Hence, the wake will recover more compared to the baseline case, as highlighted by the non-gray shaded regions downstream of the rotor. Meanwhile, the region of flow deficit will grow above and below the rotor compared to the baseline case.

Finally, for the pitch positive case, the streamwise vortical structures injected high momentum flow from above and below the rotor whilst ejecting the low momentum flow laterally outward. Hence, the wake will contract in the axial direction, increasing the availability of high energy streamwise flow for downstream rotors.

## 3 Experimental methods

### 3.1 Wind tunnel

The experiments are conducted in the open-jet facility (OJF) of TU Delft's Aerospace Engineering faculty, visualized in Figure 3. The tunnel has been used for wake studies of the same VAWT geometry in a single and two-turbine array (Huang et al. (2023c, b)) as well as for other scaled VAWT experiments (Tescione et al. (2014); LeBlanc and Ferreira (2022a)). The closed-loop wind tunnel has an octagonal outlet of dimension 2.85m $\times$ 2.85m with a contraction ratio of 3:1. The jet stream bound shear layer angle and turbulence intensity are reported as $4.7°$ and 0.5%, respectively (Lignarolo et al. (2014)). The region of uniform flow extends 6m beyond the tunnel exit, which encompasses the measurement volume for this experiment.



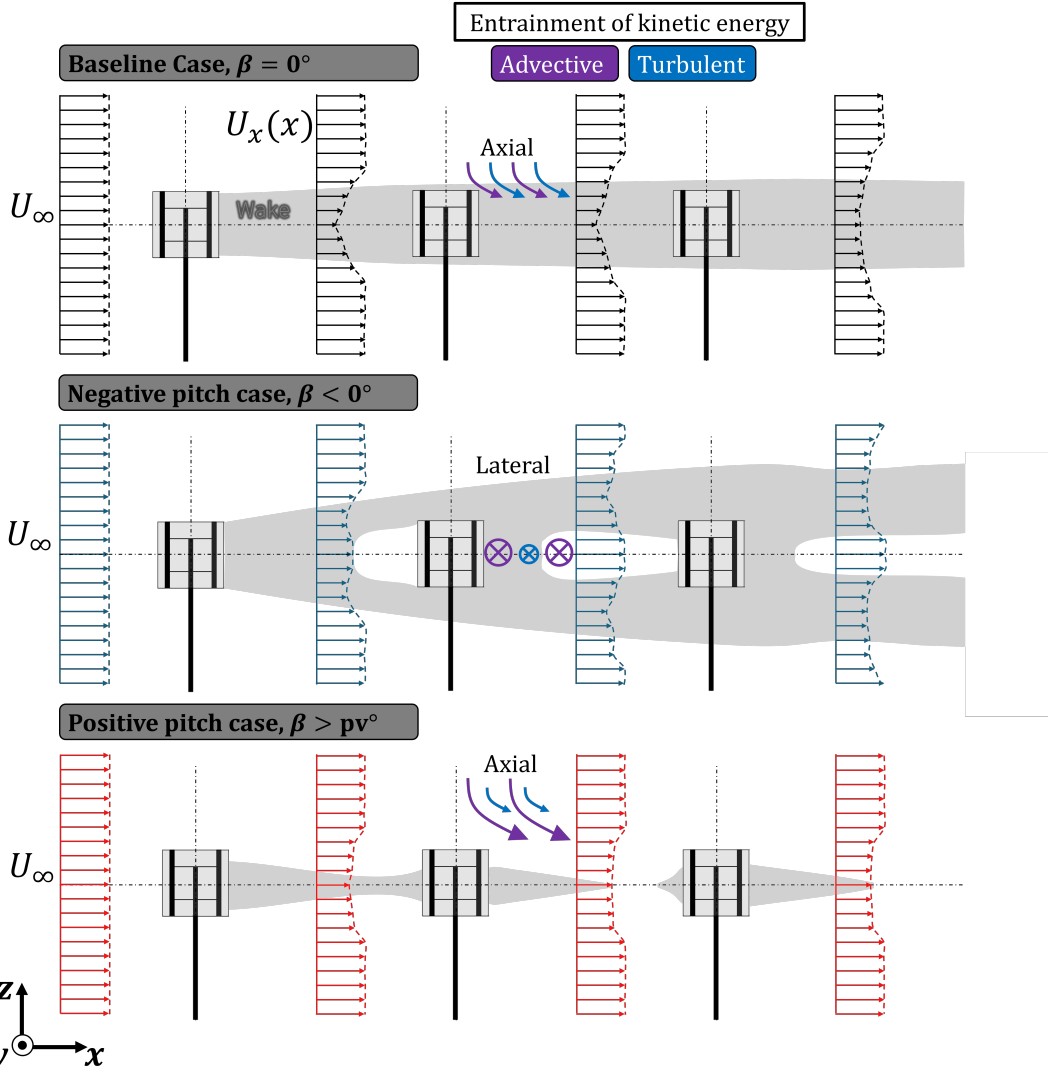

**Figure 2.** Schematic representation of the wake evolution at $xz$ plane ($y/D = 0$) of an array of three uniformly spaced inline VAWTs with laminar-uniform inflow $U_\infty$. The illustration is a simplified adaptation of numerical results obtained by Huang (2023). The gray-shaded region represents simplified wakes, while the white areas show the free stream flow. The profiles of the streamwise velocity $U_x(x)$ within the array for the three pitch cases $\beta$ are shown in black, blue, and red for the baseline, pitched out, and pitched in case, respectively. As no ground effect is included in this study, the wake structure and re-energization process will be symmetric about the symmetry plane $xy$. The advective and turbulent contributions towards the wake recovery for the three modes of operation are portrayed using purple and blue arrows, respectively. The size of the arrows corresponds to the relative contributions of each for the three cases, with the direction labeled above. In this case, the axial corresponds to the directions above and below the rotor ($z$-axis) and lateral from the sides of the rotor ($y$-axis).





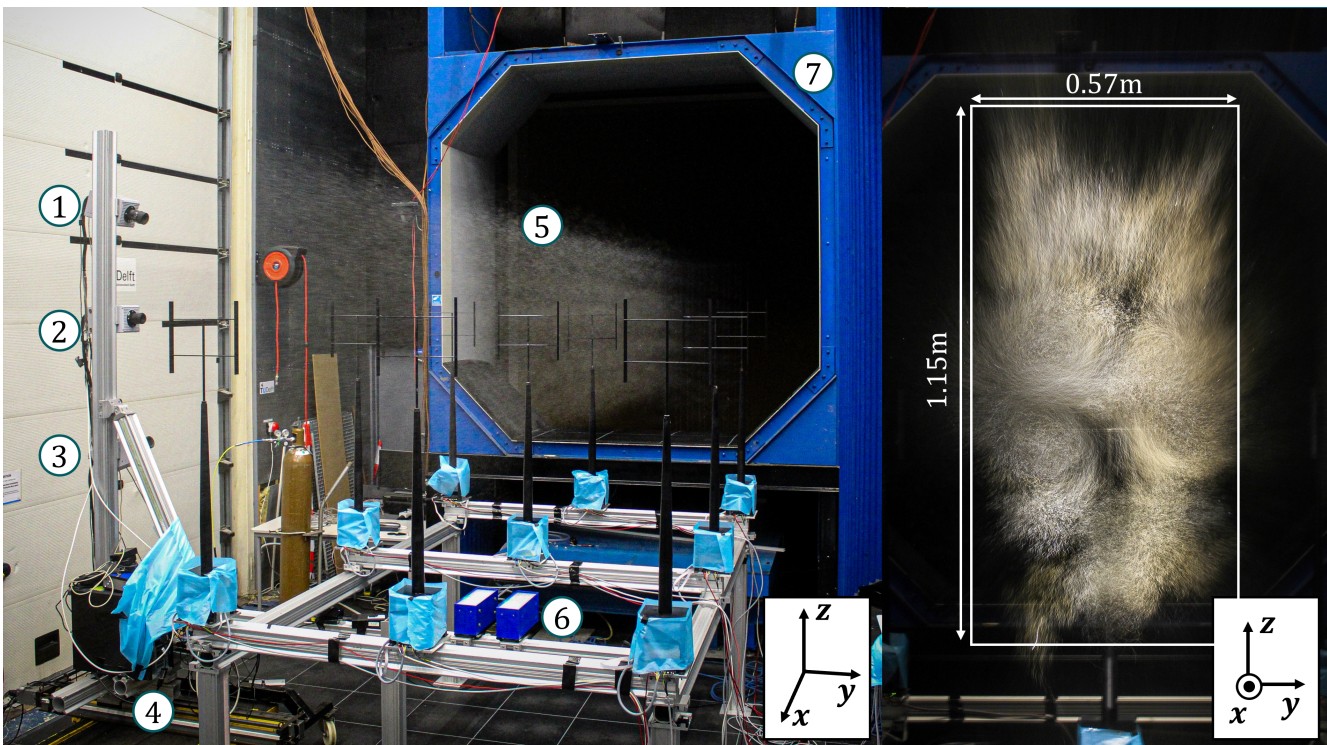

**Figure 3.** Experimental setup in the OJF. (Left) The main components are labeled and defined as follows: ① Camera 1, ② Camera 2, ③ Camera 3, ④ Traversing system, ⑤ HFSB stream tube, ⑥ LEDs, ⑦ OJF outlet. (Right) A closeup image of an illuminated measurement volume with the dimensions of the field of view is marked, showing the deformed stream tube of the HFSB. The coordinate systems for the two panels are shown on the bottom right.

## 3.2 Wind turbine model

The H-type Darrieus VAWT model is designed in-house and is an altered version of the geometry used by Huang et al. (2023b). A detailed schematic is provided in Figure 4, and illustrated in a wind farm array in Figure 3. The rotor geometry is mounted
on a rotor shaft with 10mm diameter and 1100mm height which runs through a tower. The rotor tower is 700mm tall made of milled aluminum and follows a tapered profile with base and tip diameters of 50mm and 25mm, respectively. Further, the tower houses roller bearings at the top and bottom to minimize any deflections of the shaft resulting from imbalances in the rotor. The tower is fastened to an aluminum stand, which connects to an in-house designed three-components balance. The stand houses a Maxon EC90 260W DC motor which is controlled via a ESCON 50/5 controller which rotate the rotor shaft at a constant rpm
in either clockwise or anti-clockwise direction. Prior to the wind farm flowfield measurements, the cycle averaged streamwise and lateral thrust components of the rotor are measured at different tip-speed ratios using the force balance. Furthermore, three balances are setup along the central column of the farm to measure the thrust performance of the rotors in an array, as discussed





in Section 3.4. The force balance has a maximum range of $\pm50$N and a maximum error of $\leq 0.1\%$. Further details of the design are provided by Huang et al. (2023b).

The rotor geometry consists of two straight untapered blades with a span of $S$ = 300mm fastened to the rotor shaft via two struts separated by 150mm, as shown in Figure 4. The struts have an oval-shaped profile with span, thickness, and chord of 300mm, 4mm, and 30mm, respectively. The blades are made of extruded aluminum with a NACA0012 profile with a chord $c$ = 30mm. The different fixed-pitch adjuments are realized by interchanging modular 3D-printed adapters between the blade and strut at all four connection points, shown in blue in Section 3.2. The adapters have the same chord and thickness as the struts.

In this experiment, the fixed-pitch settings of $\beta = -10°$, $0°$, and $10°$ about the mid-chord are used to allow for a qualitative comparison with the work of Huang et al. (2023b, c). Positive and negative pitch is defined as pitch inward towards the tower and outward, respectively. The rotor diameter is $D$ = 300mm. All components of the rotor are painted black to minimize the reflections during the flow measurement process, described in Section 3.3. The rotor solidity is $\sigma = NcS/A = 0.2$, where $N$ = 2 is the number of blades and $A = 0.09$ m$^2$ is the projected frontal area of the rotor. The wind tunnel blockage of a single rotor

is 0.011, considering the projected frontal area of the rotor.

### 3.3    Flow measurement system

The velocity fields are obtained through three-dimensional Lagrangian Particle Tracking. Seeding is generated using a rake placed in the settling chamber of the OJF made up of 200 nozzles, each releasing neutrally buoyant helium-filled soap bubbles (HFSB) at a rate of 30,000/sec and an average diameter of 0.4mm (Faleiros et al. (2019)). Given the rake dimensions and the

contraction ratio of the outlet, the seeded streamtube is approximately 0.57m $\times$ 1.2m, as visualized in Figure 3. The resulting seeding concentration was on average 1 HFSB/cm$^3$ (Caridi et al. (2016)). The rake has been shown to minimally impact the turbulence intensity of the incoming flow by 0.8% (Giaquinta (2018)). Given the limited size of the generated stream tube and desired lateral measurement volume, the seeding rake was translated along the width of the tunnel to adequately seed the desired measurement volume, described in Section 3.4.

The flow tracers are illuminated from below the farm using LaVision LED Flashlights 300, as shown in Figure 3, with a pulse length of 200$\mu$s. Images of the illuminated bubbles are recorded using three Photron FASTCAM SA1.1 high-speed cameras (CMOS, 1 MPx, 12 bit, 20 um pixel pitch). Image frames are acquired at a frequency of 500Hz with an image resolution of 1024px $\times$ 1024px. Cameras 1, 2, and 3 mounted lenses of 60mm, 50mm, and 60mm focal length, respectively, with set numerical aperture f/22. An individual measurement volume is approximately 0.17m$^3$, resulting in a digital image

resolution of 1.6 px/mm with a magnification factor of M = 0.03. A total of 4600 frames are captured per measurement volume, corresponding to an acquisition time of 9.2s. Given the rotational frequency of the rotor during the wake measurements of $\omega$ = 70 rad/s, approximately 102 rotor revolutions are captured. The illumination and imaging are synchronized using a LaVision PTU-X timing device. The cameras and LEDs are mounted rigidly on a traversing bed, visualized in Figure 3. This allows for the translation of the measurement volume in the streamwise and lateral directions such that several can be stitched together

without the need for re-calibrating the acquisition system, as described in Section 3.4.





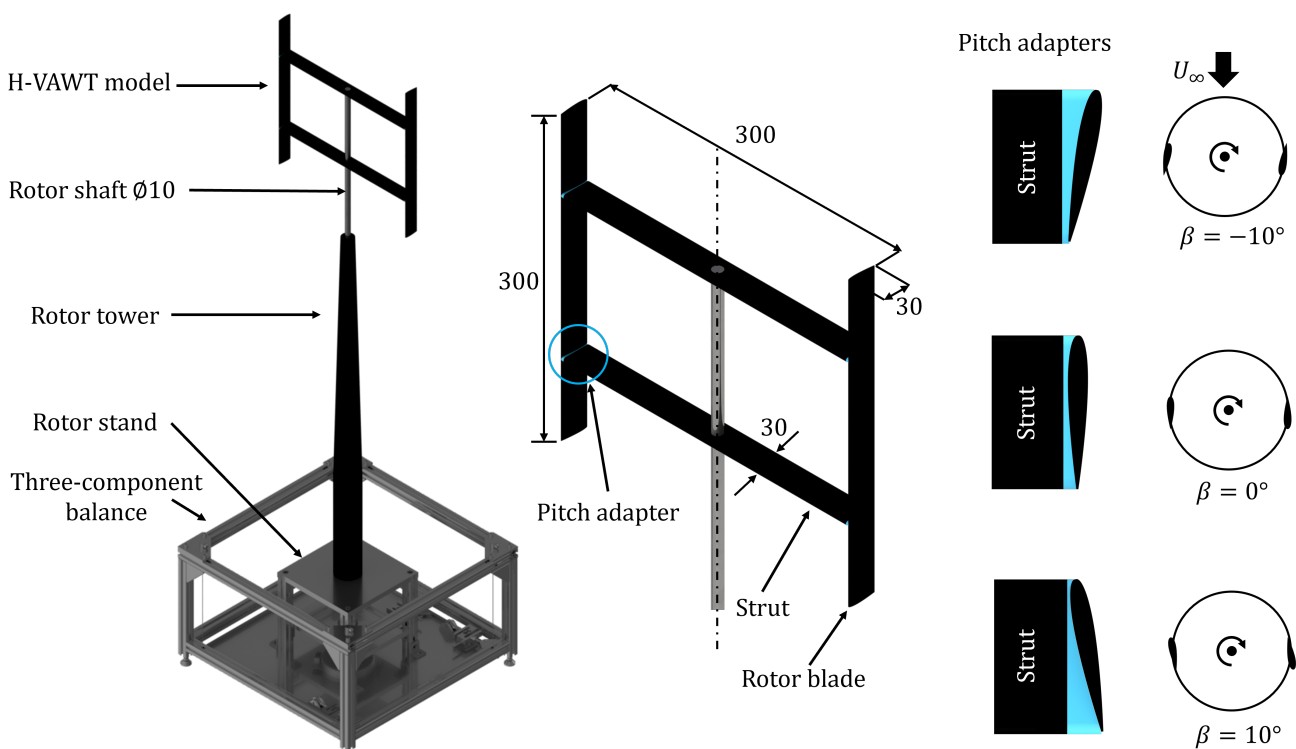

**Figure 4.** (Left) Rendered schematic of the H-type VAWT model mounted on the three-component balance, (Middle) rendering of the rotor geometry. The turquoise circle outlines one of the four pitch adapters situated at the connection point between the blades and struts, (Right) top-view rendering of the three pitch adapters (turquoise) mounted between the blade and struts, with a pitch towards and away from the tower denoted as positive and negative pitch, respectively. The top-view schematic of the rotor shows exaggerated blade sizes for clarity. The main components of the rotor are labeled, with all dimensions in mm.

The acquired frames were processed using LaVisions DaVis 10 software. First, background removal is applied by subtracting a minimum time filter with a kernel of five images. Geometric calibration of the camera positions relative to the measurement domain was performed using a calibration plate, followed by a volume self-calibration using an unperturbed flowfield recording of particles (Wieneke (2008)). The residual measurement errors fall below 0.1 pixels for all three cameras. Next, tracer trajectories were evaluated using the Shake-the-Box (STB) algorithm (Schanz et al. (2016)). The number of identified particle tracks per time-step ranges between 8000 and 15,000 depending on the specific measurement volumes in the farm and pitch case due to the strong deflection of tracers in and out of the field of view. Next, the individual STB volumes measured within the overall farm measurement volume shown in Figure 5 are stitched together with the corresponding lateral and streamwise offsets recorded using the traversing system, as described above, yielding a total measurement volume of approximately 5.3m$^2$. Finally, the binning step involves mapping particle tracks of the fully stitched volume on a Cartesian grid by spatially averaging over smaller cells (Agüera et al. (2016)). In this study, all time stamps are binned together to yield a time-averaged wake





| Pitch case | Description |
| --- | --- |
| $\beta = -10°$ | The blades of all of the rotors are pitched out |
| $\beta = 0°$ | No fixed pitch is applied to the rotors (baseline) |
| $\beta = 10°$ | The blades of all of the rotors are pitched in |

**Table 1.** Overview of the wake recovery strategy test cases

measurement. The sub-volume size is 128×128×128 voxels used, where one voxel = 80mm. Given an overlap of 75%, the final grid spacing of the velocity vectors is 19mm.

### 3.4 Case overview

The free stream wind speed is constant at $U_\infty$ = 3 m/s. The rotational frequency of all of the rotors is maintained at $\omega$ = 70 rad/s, equating to a tip-speed ratio of $\lambda = \omega R/U_\infty$ = 3.5 and diameter-based Reynolds Number of 6 $\times 10^4$.

A series of nine identical turbines are arranged in a grid beyond the exit of the OJF, as shown in Figure 3 and a top-view schematic in Figure 5. For the wake measurements, the streamwise and lateral spacing between the rows and columns are 5D (1.5m) and 3.18D (0.953m), respectively. For the load measurements of the central column described in Section 4.1, the

streamwise spacing is 4.84D (1.45m). All rotors operate at the same rotational frequency in the clockwise direction with respect to the incoming flow. The phases of the rotors are not synchronized. The origin of the coordinate system $x = y = z$ =0 is defined as the geometric center of the first rotor in the central column. Note, for the results discussed in Section 4, the measured wakes have been mirrored along the $y$−axis as if the rotors rotate contour clockwise for consistency with previous works by Bensason et al. (2023, 2024).

The three modes of operation described in Section 2 are used for this experiment. To allow for a qualitative comparison with these previous works by Huang et al. (2023b), the same fixed-pitch settings of $\beta = -10°$, $0°$, $10°$ are used. The sign convention is highlighted in Figure 4. The case with no wake mixing strategy $\beta = 0°$ will be referred to as the baseline case. The wind farm wake is measured for these three pitch cases independently, where all rotors are pitched by the same amount. Furthermore, the grid spacing between the rotors is kept constant, with no lateral offset spacing between the rotors applied. A description of

these three cases is provided in Table 1.

As discussed in Section 3.3, the full-time-averaged wake of the wind farm is measured by stitching several smaller measurement sub-volumes together using the traversing bed, which has an accuracy of ± 0.01mm. An example of the boundaries and instantaneous tracked HFSB single measurement sub-volume within the farm domain is shown in Figure 5. The overall measurement extent of the volumes at different positions within the farm for the $\beta = 10°$ case is shown in Figure 5 via colored

lines. The size of the measurement volumes varies between the different farm positions due to the strong wake deflections experienced for the vortex generator mode cases, leading to frequent areas in the flow void of any HFSB, as shown in Figure 3. To account for this, several measurement sub-volumes are captured, accompanied by frequent movement of the seeding rake in the settling chamber, and rigid camera and LED construction are used to distribute tracers more evenly and capture the full





wake. The colored lines delineate the overall measurement volume once the individual sub-volumes are stitched together, as

discussed in Section 3.3. In the inflow of the farm, two sub-volumes are measured. Between the first and second rows, nine sub-volumes are measured. Between the second and third rows, 15 sub-volumes are measured. Finally, in the outflow of the farm, 12 sub-volumes are measured. Note, due to the optical obstructions of the rotors and the nature of the measurement system, the volumes inline with the rotors are omitted from the analysis in Section 4. The individual sub-volumes are translated such that there is sufficient overlap between them within a given domain—these range between 0.7D and 1.6D in the streamwise

direction and 0.3D and 1D in the lateral direction. The final measurement domain encompasses a total streamwise distance of 17.6D, extending 2D and 5.6D in the inflow of the farm and wake behind the far, respectively. The lateral measurement ranges vary between the different sections of the farm (i.e. inflow, behind the first row, second row, etc.). To measure the inflow of the farm, it was assumed that the induction of the third row did not greatly impact the inflow, and hence, first-row rotors were removed, and the inflow of the second was measured and assumed to be that of the first row. As described in Section 2, to avoid

the ground effect, the rotors are on a frame of rigid beams, resulting in a clearance between the ground and bottom blade tip of the rotor of 2.2m (7.3D), as shown in Figure 3.

### 3.5   Flow measurement uncertainty

Given the complexity of the measurement system, several sources of uncertainty are evident regarding the quantification of velocity in the wind farm, the most important of which are related to the overall system calibration, particle detection, and

triangulation process, the signal-to-noise ratio (SNR) of the recorded images, and the quality of particle tracking and matching.

The overall system calibration and subsequent particle detection are addressed during the camera calibration phase. A geometric calibration is performed, with subsequent volumetric self-calibration (Wieneke (2008)) on a reduced sample of the particle images to remove any remaining disparities between cameras, resulting in a final pixel accuracy of 0.03px. Given the relatively low image averaged particle seeding density of between $3 \times 10^{-2}$ particle per pixel (ppp), the detection and triangu-

lation of ambiguous particles is minimized (Scarano (2012)). A threshold allowable triangulation error of 1 voxel was used to reduce ghost tracks further and improve the particle reconstruction. Pre-processed images (before background removal using the time filter described in Section 3.3) have an average intensity of 100 counts.

The standard deviations ($\sigma$) of the velocity components are computed over the total number of test samples of 4600. The magnitudes vary depending on the velocity component, pitch case, and wake location on the farm. Higher magnitudes in

standard deviation are present in the near wake along the shear layer, as concluded in previous works of Huang et al. (2023b); Bensason et al. (2024). These reach as high as $\sigma_x \approx 0.8\,\mathrm{m\,s^{-1}}$, $\sigma_y \approx 0.4\,\mathrm{m\,s^{-1}}$, and $\sigma_z \approx 0.3\,\mathrm{m\,s^{-1}}$ for the baseline pitch case $\beta = 0°$. Within the wake, these values drop to $\sigma_x \approx 0.17\,\mathrm{m\,s^{-1}}$, $\sigma_y \approx 0.15\,\mathrm{m\,s^{-1}}$, and $\sigma_z \approx 0.1\,\mathrm{m\,s^{-1}}$, respectively. For the pitched cases, the $\sigma$ remains a similar magnitude, with increases in the lateral and axial velocity components of up to $\sigma_y = \sigma_z \approx 0.6\,\mathrm{m\,s^{-1}}$ in the near wake within the shear layer. Within the induction region of the farm, the standard deviation

of the unperturbed freestream is $\sigma_x = \sigma_y = \sigma_z = 0.05\,\mathrm{m\,s^{-1}}$, and increases in magnitude near the induction region of the first rotor row to $\sigma_x = \sigma_y = \sigma_z = 0.3\,\mathrm{m\,s^{-1}}$. The corresponding velocity uncertainty $\epsilon$ is computed using Equation (1):

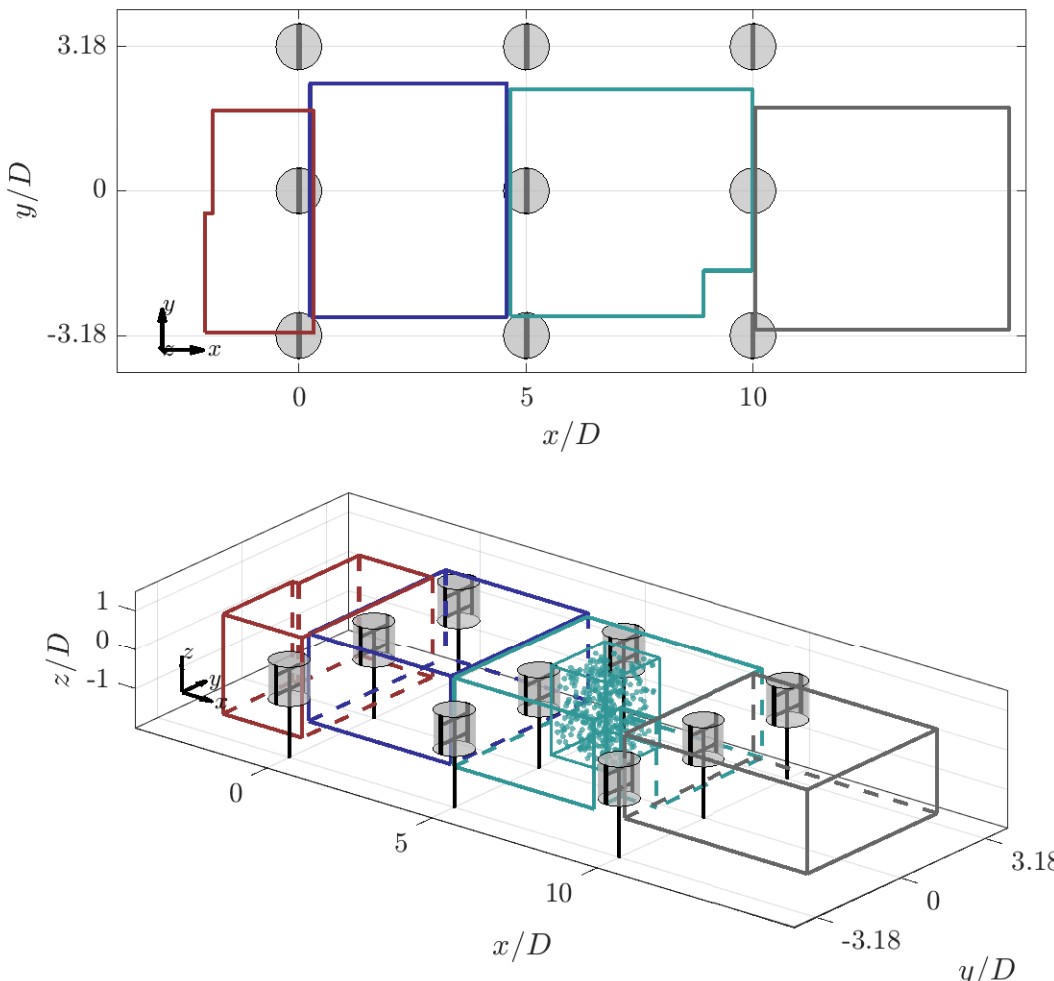

**Figure 5.** Top view (top) and isometric view (bottom) schematic of the wind farm arrangement. The wind turbines are shown at phase-locked positions $\theta = 0°$, with the surrounding gray-shaded cylinders denoting the idealized AC surfaces. The grid lines reflect the constant grid spacing between the turbines normalized by diameter $D$. The wind direction flows in the positive $x$ direction. The outlines of the overall measurement volume obtained in the inflow, between the first and second row, between the second and third row, and outflow (gray) are shown in dark red, blue, cyan, and gray, respectively. In the isometric view, a single measurement volume, as described in Section 3.3, is illustrated in the region between the second and third row. A representation of the instantaneous HFSB is provided in this volume as well.





$$\epsilon = \frac{z_a \sigma_{x,y,z}}{\sqrt{N}}, \tag{1}$$

where $N$ is the number of uncorrelated measurement samples within each bin, and $z_a$ is the coverage factor. A value of $z_a = 2$ is selected to achieve a 95% confidence interval of the normal Gaussian distribution. Given that the 4600 recordings are considered, which are time-averaged, the number of samples per bin is on the order of 900,000. Hence, the velocities reach a maximum of $\epsilon_x = 0.001\,\mathrm{m\,s^{-1}}$, which is only 0.03% of the freestream velocity $U_\infty = 3\,\mathrm{m\,s^{-1}}$. The large number of samples per bin suggests that a higher-resolution mesh would be attainable. However, the robustness of the chosen size resolves all large-scale vortices of interest while keeping the required memory to a minimum.

Furthermore, the diffraction diameter of the paticle images $d_{\mathrm{diff}}$ is calculated using Equation (2) (Raffel et al. (2018))

$$d_{\mathrm{diff}} = 2.44 \times f_\# \times (M+1) \times \lambda, \tag{2}$$

where $f_\#$ = 22 is the numerical aperture, $\lambda = 500 \times 10^{-9}$m is the wavelength of light. The resulting diffraction diameter is $d_{\mathrm{diff}} = 27.65\mu$m. Given the camera pixel size of $20\mu$m, the ratio is 1.4, which would result in minimal peak locking errors (Raffel et al. (2018)).

## 4 Results

### 4.1 Rotor thrust performance

Prior to the flowfield measurements, the thrust of an isolated rotor as a function of the tip-speed ratio is measured. These results are compared to the measurements of Huang et al. (2023c), who used the same geometry with the exception of profiled struts, as described in Section 3.2. In this work, the tip-speed ratio is varied using rotor rotational frequency $\omega$ while keeping the inflow wind speed constant at $U_\infty$ = 3m/s. In the work of Huang et al. (2023c), the rotor loads were measured at an inflow speed of $U_\infty$ = 5m/s. The streamwise and lateral thrusts, $C_{T,x}$ and $C_{T,y}$, are defined as follows:

$$C_T = \frac{T}{0.5\rho U_\infty^2 A}, \tag{3}$$

where $T$ is the measured thrust component, $\rho$ is the air density, and $A$ is the frontal area of the rotor. The measured streamwise and lateral thrust coefficients are shown in Figure 6 for the three pitch cases via solid lines and compared to published results in dashed lines.

As with the results of Huang et al. (2023c), the streamwise thrust coefficient increases and decreases when applying a positive and negative pitch, respectively. This difference is amplified as the tip-speed ratio increases. The decrease in thrust for the negative pitch case can be attributed to the reduction in load in the intrinsically more efficient upwind passage (as described in Section 2). The increased load in the downwind passage does not account for that lost upwind and is further





| Pitch case $\beta$ [°] | $C_{T,x}$ | $C_{T,y}$ |
|---|---|---|
| -10 | 0.72 | 0.386 |
| 0 | 0.79 | 0.03 |
| 10 | 0.92 | 0.10 |

**Table 2.** Measured thrust coefficients at tip speed ratio $\lambda = 3.5$

hindered by dynamic stall effects, as confirmed via blade level measurements of LeBlanc and Ferreira (2022a) at mid-span. As reported by LeBlanc and Ferreira (2022a), the blade loads increase significantly in the UW quadrant for the positive pitch case before reducing due to stall in the UL quadrant. The streamwise thrust measurements in this campaign compare well to those published by Huang et al. (2023c) for the baseline and negative pitch cases but exhibit systematically higher values for the positive pitch case on the order of 13% at a tip-speed ratio of 3. This discrepancy can likely be attributed to the difference in strut profile and pitch adapter design, as well as the difference in free stream wind speed and diameter based on Reynolds number.

The lateral thrust coefficients are positive across all pitch cases, confirming the intrinsic asymmetric loading of the VAWT in favor of the windward side. For the positive pitch mode, the lateral force increases drastically compared to the baseline case by $\Delta C_{T,y} = 0.29$ (a factor of 14.3), while the negative pitch case increases by $\Delta C_{T,y} = 0.05$ (a factor of 3.8) at a tip-speed ratio of 3.5. The lateral force in the baseline case decreases as the tip-speed ratio increases as the loading between the windward and leeward halves becomes more uniform. As in the streamwise thrust case, the positive pitch shows the largest discrepancy to results published by Huang et al. (2023c), which is again ascribed to the differences in the VAWT models and the Reynolds numbers. For the results presented, a constant tip-speed ratio of $\lambda = 3.5$ was maintained, with thrust coefficients tabulated in Table 2.

### 4.2 Streamwise vorticity system

As discussed in Section 2, the wake topology is governed by the modified trailing vorticity system of the rotor when in "vortex generator" mode. The iso-surfaces of the normalized streamwise vorticity $\omega_x D/U_\infty$ are shown for the three modes of operation in Figure 7. The values are flooded by $\omega_x D/U_\infty = \pm 1$ for the vortex generator modes, and $\omega_x D/U_\infty = \pm 0.4$ for the baseline case. Furthermore, the normalized vorticity at discrete cross-stream locations $x/D = 2.5$, 7.5, and 12.5 is presented in Figure 8. The vectors show the in-plane velocities, and the solid black line indicates the projected frontal area of the central rotor column. The dominant vortical structures stemming from the respective AC quadrants are labeled, following the same naming convention described in Section 2. The label S corresponds to structures stemming from the interfaces between the struts and blade. Given the symmetry about the $z/D = 0$ plane, only the structures on the upper half of the wake are labeled.

For the baseline case, there are no dominant vortical structures stemming from the AC quadrants behind the first rotor $x/D = 2.5$, with a presence of both upwind and downwind generated vortices on the windward and leeward sides. The UW vortex persists longer in the wake, reaching the second rotor, whereas the DW extends up to approximately $x/D = 3$, confirming

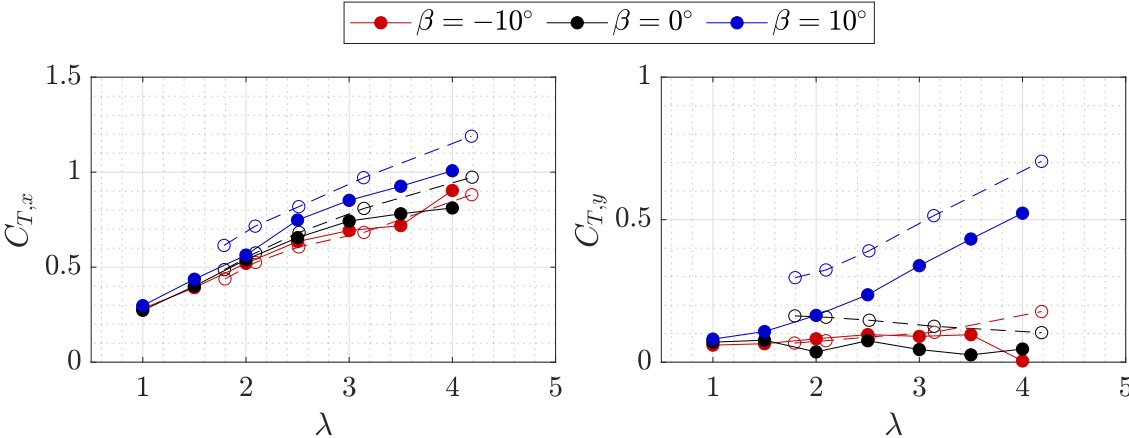

**Figure 6.** Streamwise (left) and lateral (right) thrust coefficients as a function of tip-speed ratio for the three thrust coefficients. The solid and dashed line corresponds to recorded results from this experiment and those reported by Huang et al. (2023b), respectively. All measurements of this experiment are recorded at a constant wind speed of 3m/s.

the intrinsically higher load of the upwind half. Similarly, on the leeward half of the rotor, the UL vortex persists until the second rotor in the column. The intrinsically higher load and the windward half of the cycle are further highlighted by the difference in sizes of these vortical structures as they convect through the wake. In addition to the quadrant-based vortical structures, streamwise vorticity due to the struts are present. The same structures were observed by Huang et al. (2023b) and persist both
on the windward and leeward sides. The streamwise vorticity structure behind the second rotor is analogous to that of a bluff body cylinder (Schneiders et al. (2016)), with dominant counter-clockwise and clockwise rotating structures on the windward and leeward sides, respectively. These structures induce a significant axial contraction of the wake, evidenced by the induced downwash and upwash on the upper and lower halves, respectively. As the inflow velocity of the second rotor has been reduced significantly (as evidenced in the next section Section 4.3), the effective tip-speed ratio of the rotor will increase significantly
(as the rotational frequency is kept constant) as well as the thrust coefficient. The structures persist in the wake until reaching the third rotor in the column. Behind the third rotor at $x/D = 12.5$, the same structures persist but at much lower strengths, as evidenced in the iso-surfaces.

For the positive pitch case $\beta = 10°$, the dominant vortical structures are those generated on the upwind half of the cycle, namely the UW and UL as seen at $x/D = 2.5$, as discussed in Section 2. The UW structure is larger than the UL due to
the higher load on the windward side. Due to the significant lateral force induced on the wake when pitching the blades, the structures convect towards the windward side. Those generated on the UW quadrant convect more due to their higher strength. Unlike the baseline case, the inflow for the second rotor recovers to a much higher degree due to the injection of high momentum flow from above the rotor, and hence, it exhibits a similar vortical structure to that of the first rotor. The UW-generated vortex for the rotor on the neighboring column has convected within the measurement volume, at around $y/D = -2.0$ at $x/D = 7.5$.





**Figure 7.** Iso-surfaces of the streamwise vorticity contours at $\omega_x D/U_\infty = \pm 1$ for pitch cases $\beta = \pm 10°$, and $\omega_x D/U_\infty = \pm 0.4$ for $\beta = 0°$. The rotors are shown at a $\theta = 0°$ phase, with the gray-shaded cylinders representing the idealized actuator cylinder load.





As compared to the structures of the first rotor, the UL vortices have convected further inside the projected frontal area. Finally, behind the third rotor at $x/D = 12.5$, three UW-generated vortices are present in the measurement volume, namely those generated by the second and third rotor in the column, as well as the second rotor in the neighboring column on the leeward side. As seen in the iso-surfaces, the structures generated by the third rotor are shorter than those of the second, suggesting a decrease in load of the rotor, likely attributed to the inflow of the ejected wake from the leeward side columns, as discussed in

Section 4.3.

Finally, for the negative pitch case, the dominant vortical structures in the wake are generated in the DW and DL quadrants. Behind the first rotor at $x/D = 2.5$, the DW structure remains concentrated around the upper blade tip, whilst that on the DL convects within the projected area of the rotor. The CVP on the leeward side induces a lateral injection of high-momentum flow into the wake. As in the baseline pitch case, vortices generated by the struts are apparent, albeit only on the leeward

side. Behind the second rotor at $x/D = 7.5$, the same dominant structures persist, with seemingly higher strengths. The DW-generated vortex has convected above and towards the center axis of the rotor, while the DL has spread across the entire diameter of the rotor. The CVP between the upper and lower blades on the leeward side induces a significant injection of momentum into the wake while ejecting the wake out from the windward side. Behind their rotor at $x/D = 12.5$, the same behavior persists. As in the case of the positive pitch blades, the voritcal structures do not extend as far in the wake as in the

aforementioned volumes, presumably due to the lower thrust of the third rotor and increased turbulence this further in the wind farm wake.

## 4.3    Wake topology

As discussed in Section 2, the streamwise vorticity system is an important driver of the wake topology. The normalized streamwise flow component $U_x/U_\infty$ is shown as iso-surfaces in Figure 9 for the baseline and vortex generator mode cases. The

normalized streamwise flow is flooded at values $U_x/U_\infty$ = 0.9 and 0.5, shown in orange and blue, respectively. The swept volumes of the rotors are shaded gray to represent the idealized actuator cylinder surface. Furthermore, the wake at discrete cross-stream distances $x/D$ = 2.5, 7.5, and 12.5 is shown in Figure 10 for the same cases. Finally, the normalized streamwise velocity component at the symmetry planes $y/D = 0$, and $z/D = 0$ are shown in Figure 11 and Figure 12, respectively. In-plane velocity vectors are shown for all cases. The contours at the symmetry plane $y/D = 0$ include the wake boundary at

$U_x/U_\infty = 0.6$ . The contours at the symmetry plane $z/D = 0$ show the wake center via a white line as calculated using the center of mass approach (Howland et al. (2016)), as shown in Equation (4).

$$y_c(x) = \frac{\iint y \Delta U(x,y,z) \mathrm{dydz}}{\iint \Delta U(x,y,z) \mathrm{dydz}} \tag{4}$$

where $\Delta U(x,y,z) = U_\infty - U_x(x,y,z)$. The integration is performed at discrete cross-stream planes $x/D$. As the wake is symmetric about $z/D = 0$, the integration domain is constrained to $z/D \geq 0$ such that the available data is consistent across

all pitch cases. For the baseline and negative pitch cases, the lateral domain is constrained to $-1.6 \leq y/D \leq 1.6$. In contrast, the positive pitch case is constrained to $-0.75 \leq y/D \leq 2.3$ to limit the impact of the wakes of the neighboring column on the



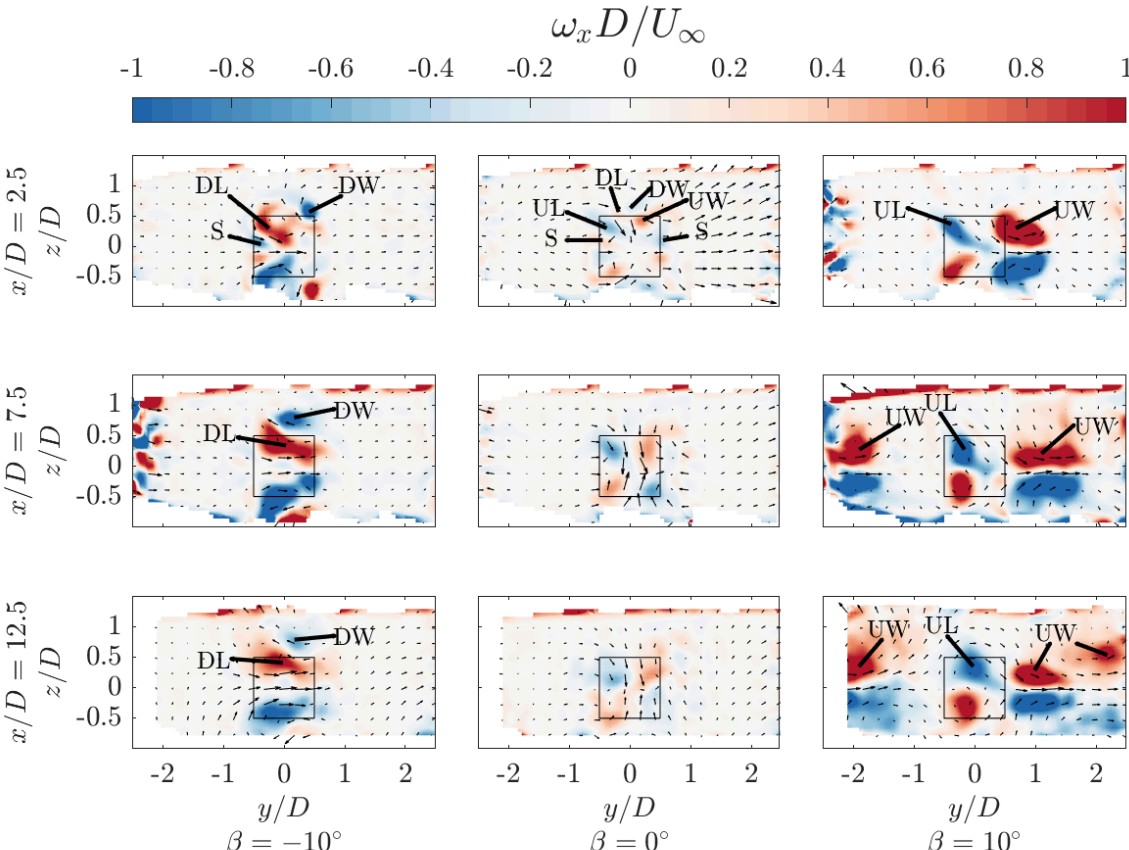

**Figure 8.** Normalized streamwise vorticity at $\omega_x D/U_\infty$ for the three pitch cases at streamwise locations $x/D$ = 2.5, 7.5, and 12.5. The black outlined square indicates the projected frontal area of the rotors, and the vectors show the in-plane velocity components. The streamwise vortical structures generated by each AC load quadrant are labeled, as described in Section 2. The label S corresponds to structures generated at the interface between the struts and blade.



computation. Beyond the third rotor's wake $x/D > 10$, the wakes between the central column and neighboring column rotors combine and become difficult to separate.

The baseline case $\beta = 0°$ exhibits the most significant flow deficit, evident by the largest blue flooded region in the iso-surfaces and blue area in the cross-stream plane $x/D = 2.5$. The cross-stream plane confirms the asymmetric load profile of the rotor, with an asymmetric wake directed towards the windward side. Similar topologies have been experimentally measured by Huang et al. (2023b); Rolin and Porté-Agel (2018). This asymmetry grows as the wake progresses downstream, as seen by the deformation of the blue-shaded wake core at $U_x = 0.5$, as well as the wake center line in Figure 12. The wake at the symmetry plane $y/D = 0$ exhibits a maximum flow deficit at approximately $x/D = 2.5$, at which point a gradual wake contraction occurs in the axial direction, evidenced by the in-plane vectors. The second and third rotors in the column are immersed in the wakes of the upstream rotors, evidenced by the extension of the orange-flooded regions within the measurement volumes. The flow at the symmetry planes has a gradually shorter wake length for the second and third rotor in the column, presumably due to decreased rotor thrust with a lower inflow velocity, as seen in Figure 12. The asymmetry in the wake grows throughout the farm, as seen in at $x/D = 7.5$ and $12.5$, with larger regions of flow deficit concentrated on the windward sides of the rotor. As there is no major lateral force component, shown in Section 4.1, the wakes of the surrounding columns do not become visible in the measurement domain until toward the end of the farm volume. As discussed in reference to the streamwise vorticity, the axial contraction of the wake at the symmetry plane $y/D = 0$ increases significantly behind the second and third rotors in the column. In contrast, the width of the wake in the lateral direction grows as the wakes superimpose.

For the positive pitch case $\beta = 10°$, a significant lateral deflection of the wake towards the windward side is evident from the iso-surfaces and cross-stream plane $x/D = 2.5$. This is consistent with the significant increase in the measured lateral force of an isolated rotor, as discussed in Section 4.1. In addition to the lateral wake deflection, there is a significant axial contraction of the wake due to the influx of high momentum from above and below the rotor. The work of Huang et al. (2023b) demonstrates a similar wake topology. The wake contraction is evident by the cross-stream planes, and the significant decrease in wake length is seen at the symmetry plane $y/D = 0$. This significant wake re-energization is also evident in the reduction in blue-shaded regions of the iso-surfaces compared to the baseline operating case. Given the recovered incoming flow for the second rotor, the wake deflection process is still effective, as seen from the symmetry plane $z/D = 0$. Also evident is the appearance of the deflected wake in the measurement domain from the neighboring column on the leeward side, starting at approximately $x/D = 2$. These wakes continue to travel laterally within the measurement volume and impinge on the third rotor in the column, as seen by the iso-surfaces and the cross-stream plane at $x/D = 12.5$. The wake center line gradually approaches $y_c/D = 0$ behind the third rotor as the wake from the neighboring column on the windward side enters the projected frontal area of the central rotors.

Finally, for the negative pitch case $\beta = -10°$, the wake is re-energized from the sides whilst ejecting low momentum flow out from above and below the rotor, evident by the elongated wake shape on the axial direction in Figure 9 and cross-stream plane $x/D = 2.5$. The injection of momentum laterally is asymmetric in favor of the leeward side and grows as the wake progresses in the farm, as seen at cross-stream planes $x/D = 7.5$ and $x/D = 12.5$. This follows the discussion of the modification and convection of the dominant downwind vortices in Section 4.2. This mode of operation is clear when observing the flow at the





symmetry plane $y/D = 0$, with gradual growth in flow deficit above and below the rotor plane, whilst flow is re-energized at the wake rotor center plane. The wake at the symmetry plane $z/D = 0$ confirms the measured increase in the lateral load of the rotor, as seen by the deflection of the wake towards the windward side. However, when considering the entire cross-section

of the wake, the wake center line is concentrated around $y_c/D = 0$. This is due to the portion of the wake that has been ejected out axially above the rotor, which connects gradually towards the leeward side, as seen in the cross-stream planes.

## 4.4 Wake recovery

The contours shown in Section 4.3 give a qualitative indication of the wake recovery mechanism of the vortex generator modes. To indicate the quantitative recovery of streamwise velocity in the wake, the normalized streamwise velocity deficits

$1 - (U_x/U_\infty)$ at symmetry plane $y/D = 0$ at cross-stream planes $x/D = 2.5, 7.5$, and $12.5$ are shown in Figure 13, each 2.5D from the central column rotors. In addition, the lateral average of the velocity within the projected regions of $-0.5 \leq y/D \leq 0.5$ (projected frontal area) and $-1.6 \leq y/D \leq 1.6$ (central column grid size) are presented, denoted as $y/D \pm 0.5$ and $y/D \pm 1.6$, respectively. Additionally, the normalized axial velocity component $U_z/U_\infty$ for each case is presented.

The streamwise velocity distributions about the symmetry plane $y/D = 0$ confirm the vortex generator mode of operations

discussed in Section 2. For the baseline case, there is a large deficit behind the first rotor in the column, reaching $1 - (U_x/U_\infty)$ $= 1.1$, suggesting a small region of backflow in the wake, visible in Figure 11. The deficits are lower behind the second and third rotors as the rotors produce less thrust, yielding shorter wake lengths, as discussed in Section 4.3. The maximum deficits behind the second and third rotor are $1 - (U_x/U_\infty) = 0.63$ and $1 - (U_x/U_\infty) = 0.6$, respectively. The profiles for all three cross-stream locations are relatively symmetric about the $z/D = 0$ plane with minor differences due to the rotor tower, consistent

with the discussion in Section 2. The axial velocity behind the first rotor at $x/D = 2.5$ is low in magnitude and reaches a local minimum of $U_z/U_\infty = -0.04$ near the rotor tips, suggesting a wake expansion. Behind the second and third rotors, the profile approaches that of the positive pitch case, suggesting a significant wake contraction, as discussed in Section 4.3, with minima of $U_z/U_\infty = -0.16$ in the upper half.

For the vortex generator mode with negative pitch $\beta = -10°$, the flow behind the first rotor is substantially recovered

compared to the baseline case within the range $-0.5 \leq z/D \leq 0.3$, reaching a maximum deficit of $1 - (U_x/U_\infty) = 0.26$. As the wake is ejected out axially above and below the rotor, as suggested by the $U_z/U_\infty$ profile, the flow deficits above and below the projected frontal area of the rotor are greater than the baseline case, reaching $1 - (U_x/U_\infty) = 0.5$. The wake region extends up to a height of $z/D = 0.9$ compared to the baseline case of $z/D = 0.6$, with a maximum upwash of $U_z/U_\infty = 0.1$ above the rotor plane. Behind the second rotor, the maximum deficit decreases to $1 - (U_x/U_\infty) = 0.12$. The wake region continues

to expand above and below the rotor, encompassing a larger range above $z/D = 1.3$, while reaching a lower maximum deficit of $1 - (U_x/U_\infty) = 0.34$. Behind the third rotor, the wake region expands even further above the area of the rotor and reaches a maximum deficit within the bounds of the rotor of $1 - (U_x/U_\infty) = 0.23$. The maximum upwash decreases between the cross-stream planes $x/D = 7.5$ and $12.5$, from $U_z/U_\infty = 0.16$ to $0.1$, respectively. This is due to the decrease in vortex strength, as seen in Figure 8.





**Figure 9.** Iso-surfaces of the normalized streamwise velocity $U_x/U_\infty = 0.5$ and 0.9 in blue and orange, respectively, for the three pitch cases. The rotors are shown at a phase-locked position $\theta = 0°$, with the gray-shaded cylinders representing the idealized actuator cylinder load.

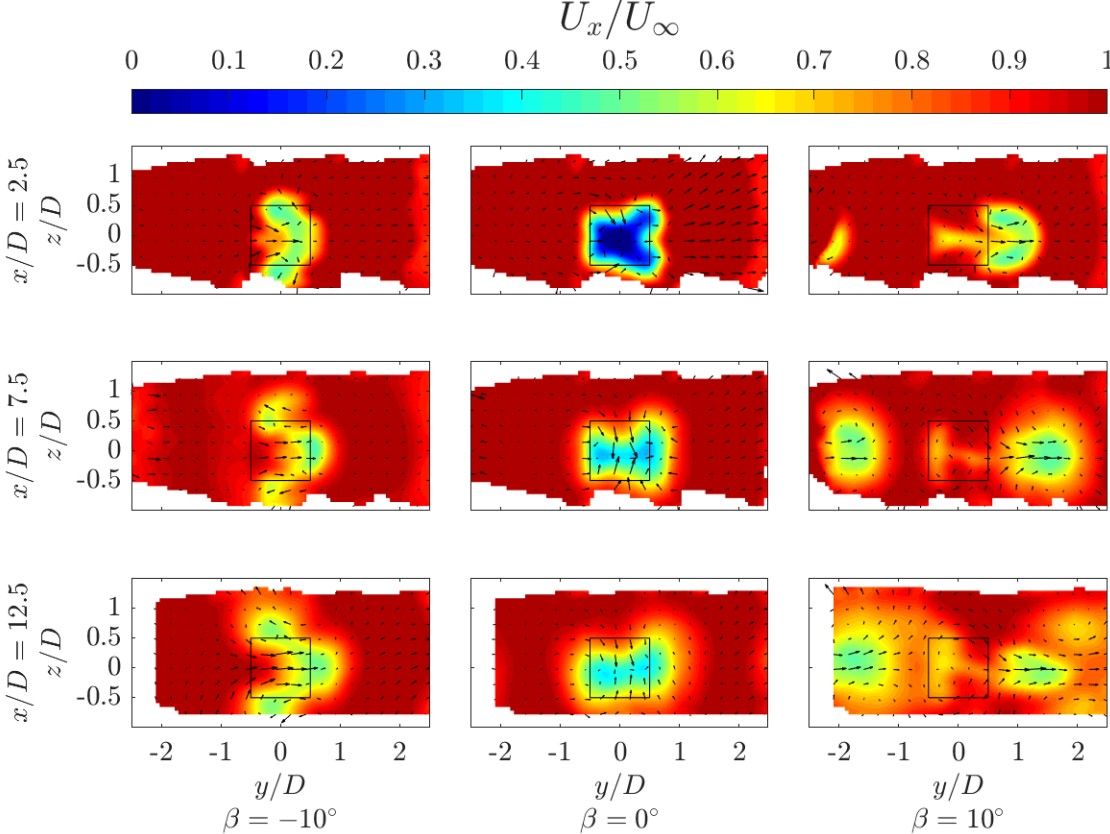

**Figure 10.** Normalized streamwise velocity $U_x/U_\infty$ for the three pitch cases at stream-wise locations $x/D$ = 2.5, 7.5, and 12.5. The black outlined square denotes the projected frontal area of the rotor, and the vectors show the in-plane velocity components.

Finally, for the vortex generator mode with positive pitch $\beta = 10°$, the wake is axially contracted whilst ejected out laterally. Hence, a significant reduction in the streamwise flow deficit behind the first rotor is evident, with a maximum deficit of $1 - (U_x/U_\infty) = 0.32$. The wake region has also contracted, reaching a height of $z/D = 0.3$, suggesting freestream flow within the projected frontal of the rotor, highlighted in Figure 11. The downwash on the upper half of the rotor reaches a local minimum of $U_z/U_\infty = -0.2$. Behind the second rotor, the maximum wake deficit decreases to $1 - (U_x/U_\infty) = 0.17$ whilst

maintaining a similar wake width. The local minima of the downwash remain similar, with $U_z/U_\infty = -0.18$. The location of this minima shifts due to the larger deflection of the wake behind the second rotor, as seen in Figure 12. The symmetry above and below the rotor begins to deviate behind the third rotor as the wakes of the rotors on the neighboring column enter the measurement volume, as shown in Figure 10. Similarly, the maximum magnitudes of the downwash and upwash on the upper and lower halves of the wake, presumably due to the different lateral locations of the UL vortices, as shown in Figure 8.

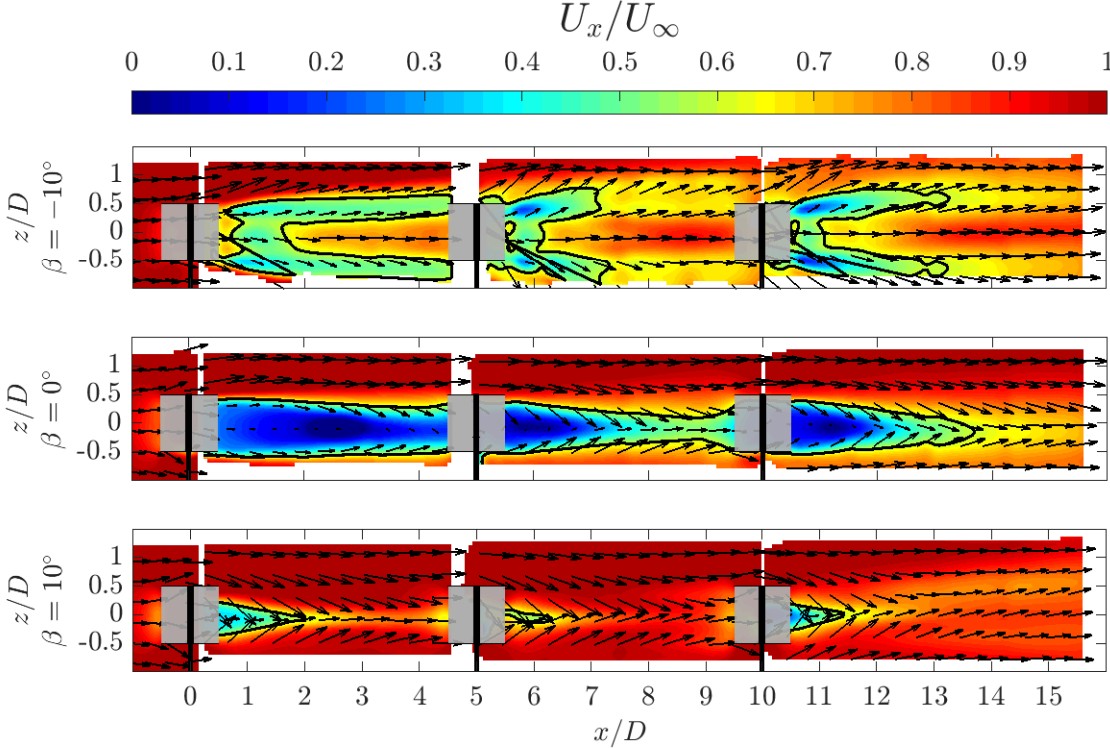

**Figure 11.** Normalized streamwise velocity $U_x/U_\infty$ for the three pitch cases at the constant lateral plane $y/D = 0$. The vectors show the in-plane velocity components. The thick black line denotes the wake boundary at $U_x/U_\infty = 0.6$. The grey shaded squares represent the idealized actuator load surface, and the black volumes show the rotor at a phase of $\theta = 0°$.

When taking the lateral average within the projected frontal area of the rotor $-0.5 \leq y/D \leq 0.5$, the trend is similar to the non-averaged profile. The baseline case exhibits the highest flow deficit across all three measurement planes, with similar maximum deficits behind the second and third rotors. For the negative pitching case $\beta = -10°$, the mean streamwise velocity recovery within the projected frontal area of the rotor is higher than the baseline case across the three measurement planes. As described in the aforementioned case, the wake is ejected out axially above and below the rotor, leading to higher average

deficits within the extended lateral domain beyond $z/D \geq 0.5$ compared with the baseline case. The profile does not exhibit as notable peaks in streamwise flow deficit above and below the rotor as with the profile from the symmetry plane $y/D = 0$. Finally, for the positive pitch case $\beta = 10°$, the trend is similar to that on the symmetry plane. The axial flow profiles also maintain a similar trend to the non-averaged case. However, a clear difference appears for the positive pitch case, where the maximum magnitude decreases more rapidly between the cross-stream planes, from $U_z/U_\infty = 0.13$ to $0.02$ at $x/D = 2.5$ and

12.5, respectively. This can be attributed to the gradual convection of the UL vortices within the projected volume of the rotors.



**Figure 12.** Normalized streamwise velocity $U_x/U_\infty$ for the three pitch cases at the constant axial plane $z/D = 0$. The vectors show the in-plane velocity components. The white markers show the wake center as computed using Equation (4).The grey shaded squares represent the idealized actuator load surface, and the black volumes show the rotor at a phase of $\theta = 0°$.





When extending the domain for the lateral average to $-1.6 \leq y/D \leq 1.6$, the relationships between the three modes of operation begin to deviate from the aforementioned cases. In general, the deficits increase downstream as the wakes superimpose and the deflected wakes from the neighboring columns enter the extended domain. Behind the first rotor, the baseline case $\beta = 0°$ exhibits the highest deficit with $< 1 - (U_x/U_\infty) >= 0.3$. However, due to the extended domain in the lateral direction,

the negative pitch case $\beta = -10°$ exhibits the highest wake recovery with $< 1 - (U_x/U_\infty) >= 0.1$ compared to the positive pitch case $\beta = 10°$ with $< 1 - (U_x/U_\infty) >= 0.19$. This is because the ejected wake of the positive pitch mode of operation falls within the extended lateral range, whereas the axially ejected wake of the negative pitch mode falls outside the integration window. Behind the second rotor, the relationship between the three modes of operation remains the same, with higher maximum streamwise deficits. The maximum deficits between the positive pitch case and baseline become more similar as

the wakes from the surrounding columns deflect within the interrogation volume. Finally, behind the third rotor, the maximum flow deficits for the baseline and positive pitch case are very close in magnitude, with $< 1 - (U_x/U_\infty) >= 0.33$ and $< 1 - (U_x/U_\infty) >= 0.31$, respectively. This is attributed to the deflected wakes from the neighboring column for the positive pitch case, seen in Figure 12.

### 4.5 Available power

The flowfield contours shown in Section 4.3 visually confirm the injection of high momentum flow into the wake for both vortex generator modes given the availability of freestream inflow for the second and third rotors in the central column. To further highlight the efficacy of these modes of operation on the replenishment of kinetic energy in the wake, the potentially available power in the wake is quantified at a selected cross-stream location using the coefficient of available power $f_{AP}$. A similar analysis has been performed by Huang et al. (2023b); Bossuyt et al. (2021); Hezaveh and Bou-Zeid (2018) to highlight

the replenishment of high momentum flow inside a selected domain, and is computed as:

$$f_{AP}(x, y_0, z_0) = \iint\limits_S U_x^3(x,y,z)/U_\infty^3 \mathrm{dzdy}, \tag{5}$$

where $U_x^3/U_\infty^3$ is the coefficient of available power at a select cross-stream plane $x$. The surface integral is computed within a defined domain ($S$), which is centered at $y_0, z_0$ in the lateral and axial directions, respectively. The available power coefficients throughout the central column of the wind farm are shown in Figure 14 for the three modes of operation. The average available

power is computed within the projected frontal area of the central column rotors $y_0/D = z_0/D = 0$. The vertical dashed lines indicate the boundaries of the rotor-swept areas for the central column. Note that, as the powers of the rotors in the farm were not measured, the presented curves are not scaled to account for the extracted power from the flow of the rotors within the central column.

For the baseline case within the projected frontal area of the rotor, the available power does not surpass 20% between the

first and second rows. The AP reaches a minimum of $< U_x^3/U_\infty^3 > = 0.06$ at $x/D = 1.4$ before steadily recovering due to the inherent diffusion of the wake. Between the second and third row, the AP recovers much faster, consistent with the shorter wake length observed in Figure 11 and Figure 12, as well as the lower recorded thrust coefficient of the second rotor. The AP



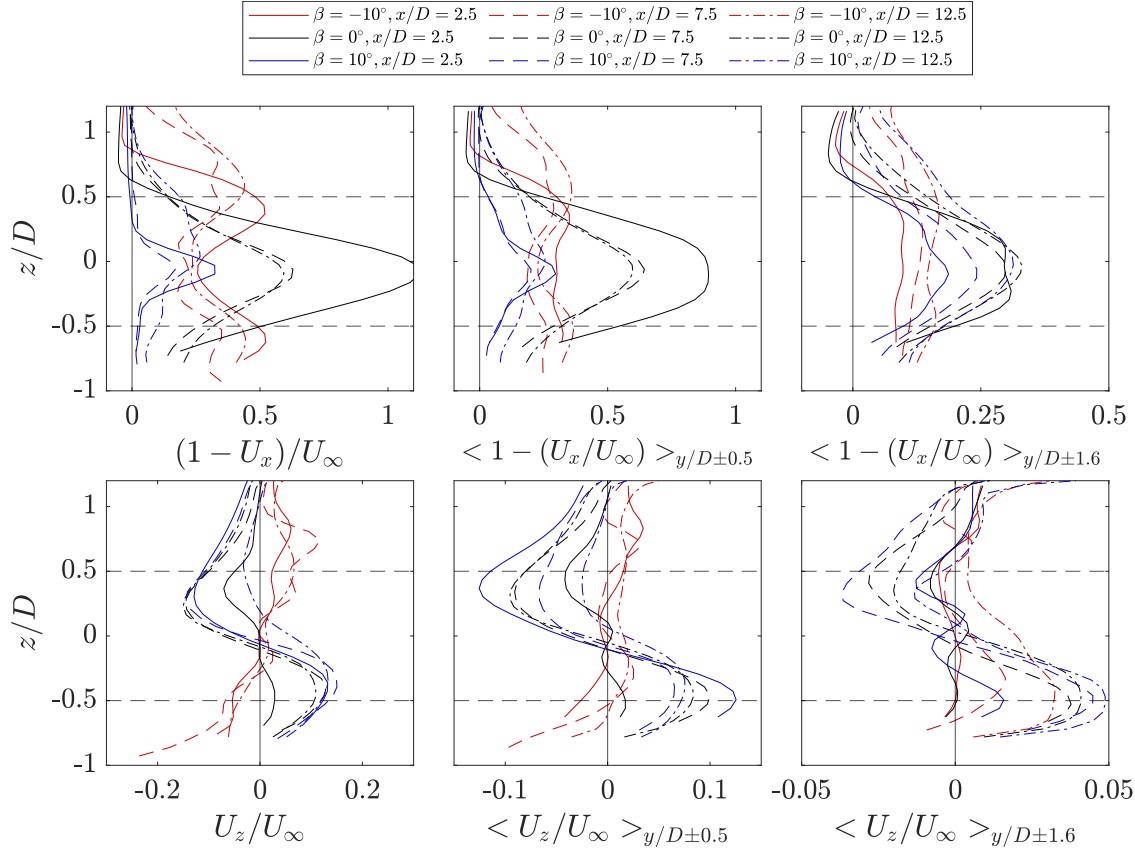

**Figure 13.** Normalized streamwise deficit $1 - (U_x/U_\infty)$ and axial $U_z/U_\infty$ velocity profiles at discrete cross-stream measurement planes $x/D$ = 2.5, 7.5, and 12.5 for the vortex generator and baseline modes of operations. (Left column) the normalized velocity profiles about the $y/D = 0$ symmetry plane. (Middle column) lateral-averaged streamwise velocity profiles within the projected frontal area of the rotor $-0.5 \leq y/D \leq 0.5$. (Right column) lateral-averaged velocity profiles within lateral domain $-1.6 \leq y/D \leq 1.6$. The horizontal dashed lines indicate the upper and lower bounds of the rotor projected frontal area.



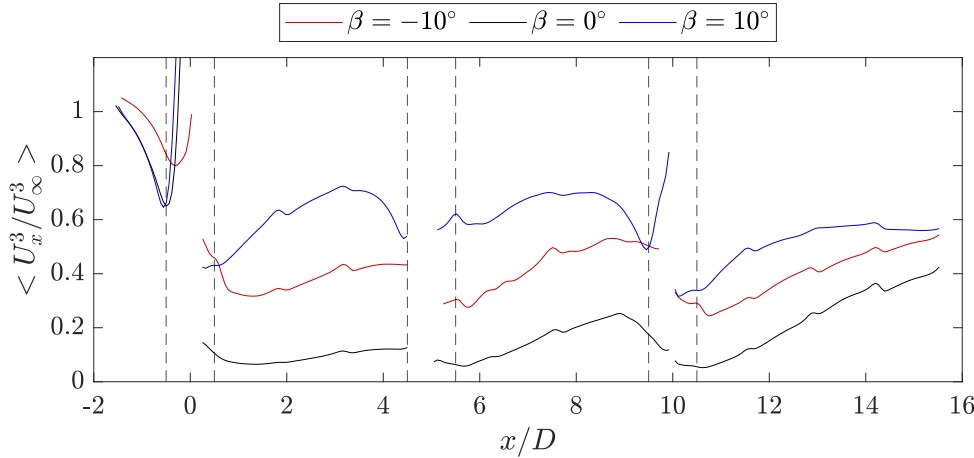

**Figure 14.** Mean available power $< U_x^3/U_\infty^3 >$ as calculated using Equation (5) for the three pitch cases over streamwise distance $x/D$ within the projected frontal area of the central rotor column $-0.5 < z/D, y/D < 0.5$. The vertical black dashed lines correspond to the boundaries of the swept volumes of the central column rotors.

reaches a maximum of 0.25 at $x/D = 9$ before decreasing upstream of the third rotor in the column due to its induction. Finally, downstream of the third rotor, the AP increases at a similar rate to the previous case, reaching a recorded maximum of 0.42 at

$x/D = 15.5$.

For both vortex generator control strategies, the available power in the wake is higher than the baseline case within the projected frontal area of the rotor. The case where momentum is injected from above the rotor $\beta = 10°$ exhibits higher AP across all measurement domains compared to the alternative case of $\beta = -10°$. Directly behind the first rotor the negative pitch case reaches minimum AP of 0.32 at $x/D = 1.3$ before gradually increasing. The positive pitch case reaches a maximum AP

of 0.72 at $x/D = 3.2$ before a steep drop due to the amplified induction of the second rotor. This drop due to induction is not as visible in the negative pitch case as the load shifted from the upwind quadrant. Between the second and third rotors, the AP is higher directly downstream of the rotor for the positive pitch case compared to the aforementioned volume. However, a comparable maximum AP is reached of 0.7 at a similar distance from the rotor of $x/D = 8.4$ before dropping due to the induction of the third rotor. Contrarily, the negative pitch case reaches a higher maximum compared to the previous volume

of AP = 0.53 at $x/D = 8.7$. Finally, behind the third rotor, the positive pitch case shows overall lower AP compared to the previous volumes. Inspection of the flow fields in Section 4.3 shed light on the reason, where the deflected wakes from the side columns have entered the domain directly downstream of the third rotor of the central column. Conversely, the negative pitch case shows a similar rate of increase in AP compared to the aforementioned volume as the wakes from side columns deflect far less compared to the positive pitch case, reaching a comparable AP of 0.54 at the final measurement location of $x/D = 15.5$.

To shed light on the potential optimal placement of further downstream turbines with the freedom to have lateral offsets, the AP coefficient is computed within the projected frontal area of the rotor centered at different lateral offsets to $-1.5 \le y_0 \le 1.5$





overall all streamwise measurement planes. These are presented as a contour plot in Figure 15 for the three modes of operation, where the solid black line indicates where $f_{\mathrm{AP}} = 0.7$, i.e. a 70% recovery of the maximum AP from the streamwise flow. Furthermore, the sliding window AP are extracted at three discrete streamwise locations $x/D$ = 2.5, 7.5, and 12.5 for three

modes operations and shown in Figure 16.

For the baseline case, the wake is concentrated within the projected frontal area of the rotor, with beyond 70% AP on the windward and leeward sides of the rotor. The AP distribution is slightly asymmetric, with more of a deficit on the windward side of the rotor. This is consistent with the asymmetric loading of the rotor described in Section 2. Behind the second and third rows, the depleted energy regions grow on either side of the rotor as the wake superimposes on each other, as seen in

Figure 16 by the gradually flattening curves and diminishing maximum AP deficits, increasing from 0.08 at $x/D$ = 2.5 to 0.2 at $x/D$ = 12.5. For the negative pitch case where momentum is entrained through the sides of the rotor, the region where AP falls below 70% is narrower as the wake is ejected out of the volume axially. Consistent with the increase in the lateral load shown in Section 4.1, the wake deflects towards the windward side, yielding lower AP potential for rotors positions in that direction. The AP is higher than that of the baseline case across all lateral positions, as shown in Figure 16. Finally, for the

positive pitch case, the regions of AP below 70% are deflected towards the windward side of the rotor, yielding a large region in the leeward side where a potential rotor can experience close to 100% of the AP. Within the range of $-0.6 \leq y_0/D \leq 0.4$, the AP of the positive pitch case is higher than the other two modes of operation, beyond which yields lower AP on the windward side. Behind the second rotor, the wake of the rotor in the first row persists on the windward side. On the leeward side, the region of high potential AP is smaller due to the wake of the rotors in the neighboring column. Finally, behind the third rotor,

the region where the AP surpasses 70% is again reduced between the wakes of the third rotor and neighboring columns. This mode of wake recovery yields higher AP at $x/D$ = 12.5 in a narrow lateral spacing window of $-0.1 \leq y_0/D \leq 0.9$.

### 4.6 Surface momentum flux

The discussion in Section 4.5 highlighted the potential for increased energy density of a VAWT farm using a wake recovery strategy. To help quantify the process of wake recovery, a volume analysis of the energy equation is adopted. Similar techniques

have been used by Hezaveh and Bou-Zeid (2018); Cortina et al. (2016), while works of Huang et al. (2023b); Bossuyt et al. (2021) used the momentum equation terms to highlight the recovery processes of individual VAWTs and HAWTs. The mean kinetic energy (MKE) equation can be derived by from the momentum equation and is formulated as (Hezaveh and Bou-Zeid (2018)):

$$\frac{\partial\left(\frac{1}{2}U_x^3\right)}{\partial x} = -\frac{\partial\left(\frac{1}{2}U_y U_x^2\right)}{\partial y} - \frac{\partial\left(\frac{1}{2}U_z U_x^2\right)}{\partial z} - U_x\frac{\partial\left(\overline{U_x' U_x'}\right)}{\partial x} - U_x\frac{\partial\left(\overline{U_x' U_y'}\right)}{\partial y} - U_x\frac{\partial\left(\overline{U_x' U_z'}\right)}{\partial z} \tag{6}$$

The expressions denoted with primes are the fluctuations of the temporally averaged velocity components in each direction with subscripts $x, y, z$. As in the work of Huang et al. (2023b); Bossuyt et al. (2021), the equation is re-arranged to equate the advective and Reynolds stresses contributions towards the streamwise energy recovery term. The viscous forces are neglected due to the sufficiently high diameter-based Reynolds number. The pressure term is omitted as it was not measured. Furthermore,



**Figure 15.** Available power as calculated using Equation (5) for the three pitch cases over all streamwise distances $x/D$ and lateral offsets $y_0/D$ ranging from -1.5 to 1.5. The gray areas indicate the top-view areas of the rotor and the idealized actuator surface. The black contour lines show where $f_{AP} = 0.7$, i.e., 70% of the maximum available power provided by the streamwise flow.

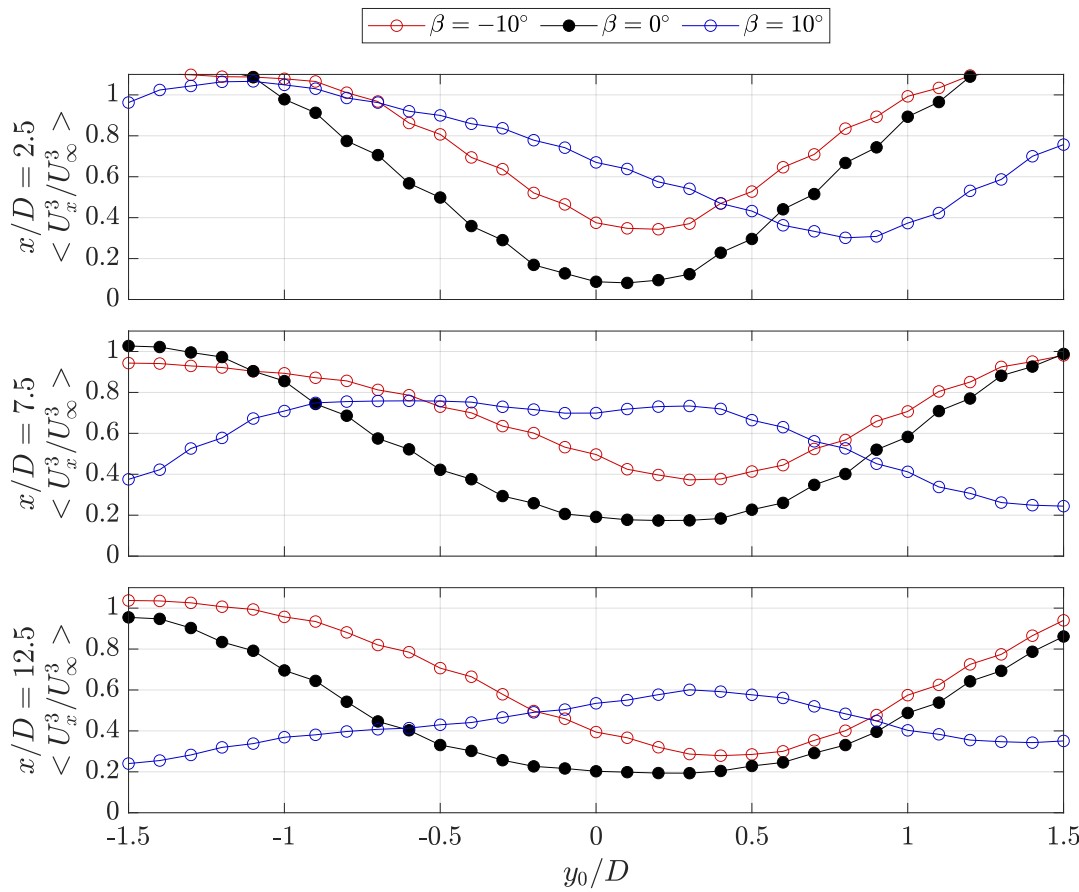

**Figure 16.** Available power as calculated using Equation (5) for the three pitch cases at discrete streamwise distance $x/D$ of 2.5, 7.5, and 12. The AP is computed within the projected frontal area of the rotor $-0.5 < z/D, y/D < 0.5$ with a lateral center $y_0$ with the discrete measurement plane.




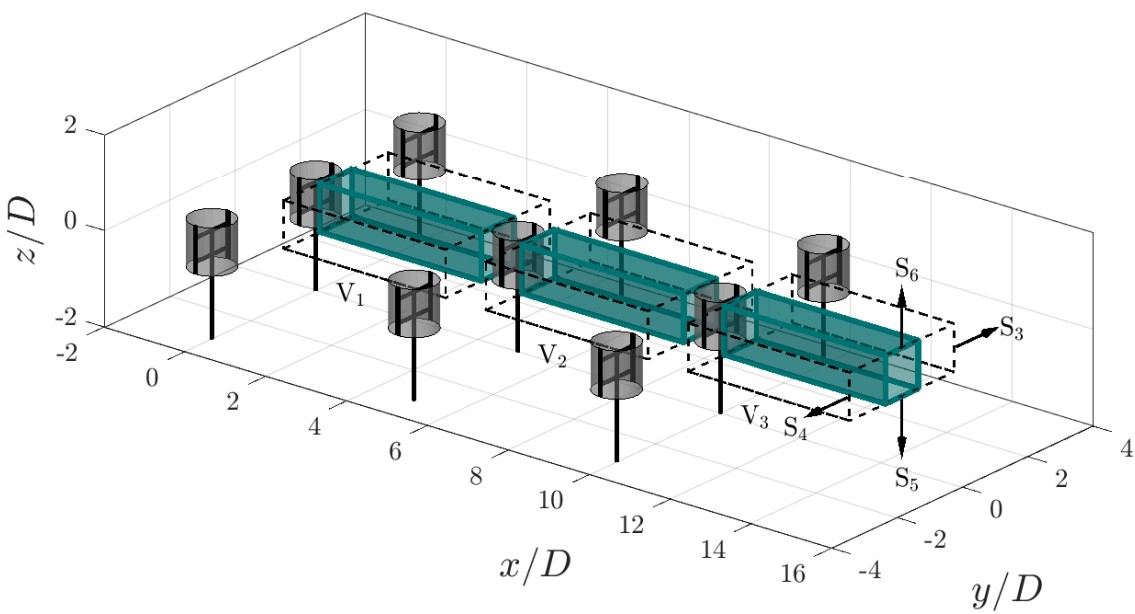

**Figure 17.** Schematic the wind farm control volumes, denoted as $V_1$, $V_2$, and $V_3$, corresponding to the flow behind the first, second, and third rotor of the central column, respectively. The shaded regions are the volumes within the projected areas of the rotors of the central column ($-0.5 \leq y/D \leq 0.5$), while the dashed black lines indicate the edges of the central cells of the farm grid ($-1.6 \leq y/D \leq 1.6$). The fluxes are evaluated through the lateral and axial surfaces $S_3$, $S_4$, $S_5$, and $S_6$, respectively. The gray shaded cylinder indicates the idealized AC surface of the rotors, which are shown at a phase-locked position of $\theta = 0°$.

terms regarding the Coriolis, thermal buoyancy, and turbine forcing as considered in the work of Cortina et al. (2016) are
exempted from focusing on the driving advective process of the streamwise vorticity system.

To investigate the relative contributions of the aforementioned terms towards wake re-energizing within the wind farm, a control volume analysis is adopted, as in the works of Cortina et al. (2016); Hezaveh and Bou-Zeid (2018). The proposed control volume is illustrated in Figure 17. The volume encompasses the entire measurement domain in the streamwise direction, ranging between $-2 \leq x/D \leq 16$. The upper and lower surfaces of the control volume are taken as the heights of the upper
and lower blade tips $-0.5 \leq z/D \leq 0.5$. Finally, the lateral size is taken as the diameter of the wind turbines, with $-0.5 \leq y/D \leq 0.5$. Given the limited measurement domain in the axial direction, this ensures an equal surface area considered for the flux on the axial and lateral surfaces. As the mean kinetic energy along the streamwise direction is the most critical towards turbine performance, the focus is placed on the flux of energy lateral and axial surfaces, while the inflow and outflow surfaces ($S_1$ and $S_2$, respectively) are omitted from the subsequent analysis.
Following the procedure outlined by Cortina et al. (2016), the volume integral of the terms in Equation (6) can be simplified by applying the divergence theorem to rewrite the terms to a surface integral form. This is simplified further by considering





that velocities normal to the surfaces do not contribute to the flux. Hence, the fluxes through the normals of the control surfaces $S_{3,4,5,6}$ can be expressed as:

$$\text{Surface 3}: \iint\limits_{S_3} U_x \left(0.5 U_y U_x + U_x' U_y'\right) dS_3 \tag{7}$$


$$\text{Surface 4}: -\iint\limits_{S_4} U_x \left(0.5 U_y U_x + U_x' U_y'\right) dS_4 \tag{8}$$

$$\text{Surface 5}: -\iint\limits_{S_5} U_x \left(0.5 U_z U_x + U_x' U_z'\right) dS_5 \tag{9}$$

$$\text{Surface 6}: \iint\limits_{S_6} U_x \left(0.5 U_z U_x + U_x' U_z'\right) dS_6 \tag{10}$$

In this case, surfaces $S_4$ and $S_5$ are given negative signs such that flux entering and leaving the volume is denoted as positive and negative, respectively.

The advective and Reynolds stress contributions towards the fluxes through each surface as denoted in Equation (7) are presented in Figure 18 and Figure 19, respectively. The line integrated fluxes across the denoted axial and lateral surfaces are computed as a function of the streamwise position $x/D$ for the three modes of operation within the farm grid lateral boundaries of $-0.5 \leq y/D \leq 0.5$. The values are normalized by the $D/U_\infty^3 L$, where $L$ is the integration length in the lateral direction, in this case, $L = D$.

For the baseline pitch case, a minimal flux across the lateral boundaries of the volume is visible, consistent with the flowfield visualized in Section 4.3. The surfaces show a degree of symmetry consistent with the wake expansion shown in Section 4.3. For the negative pitch case, the trend in flux on $S_3$ steadily decreases within a volume when traveling downstream, consistent with the lateral deflection of the wake core on the windward side of the rotor. Correspondingly, the flux on the leeward side of the rotor ($S_4$) is higher in magnitude as high momentum freestream flow is injected into the wake. This reaches increasing

maximum magnitudes of 0.05, 0.06, and 0.08 in Volumes 1, 2, and 3, respectively. This increase in flux is apparent when considering Figure 8, where an increasing asymmetric lateral inflow is shown as the CVP convect towards the upper and lower boundaries of the rotor. The positive pitch case exhibits the highest magnitude in lateral advection on $S_3$, reaching maximums of 0.07, 0.1, and 0.08 in Volumes 1, 2, and 3, respectively. This high lateral flux component is attributed to the ejection of the wake on the windward side of the rotor. Meanwhile, on the $S_4$, there is an injection of momentum, which grows

in magnitude between the first and second volumes. Leading up to the third rotor in the column, the momentum injection decreases significantly due to the wake of the neighboring column on the leeward side.

Given the uniform inflow case of this study, the flow remains symmetric about the plane $z/D = 0$, as noted in the previous analysis in Section 4.3 and Section 4.2. Hence, the fluxes about $S_5$ and $S_6$ are also the same. Minor differences between





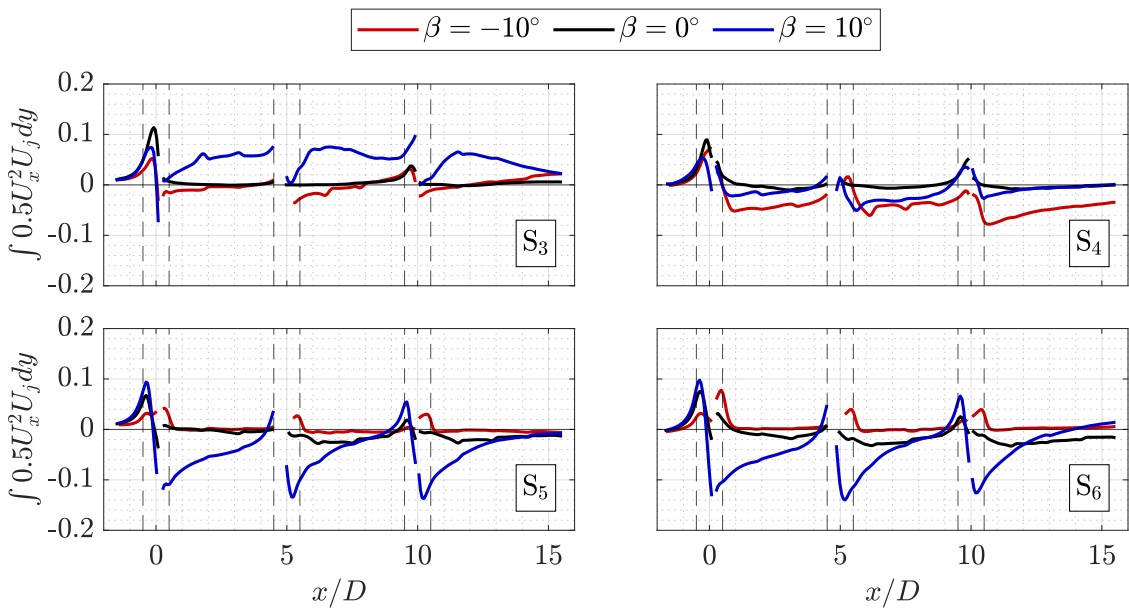

**Figure 18.** Line integrated advective flux terms as a function of downstream position $x/D$ through the surfaces $j = S_{3,4,5,6}$, as noted in Figure 17 for the three modes of operation. The flux terms are integrated within the lateral boundaries $-0.5 \leq y/D \leq 0.5$. All values fluxes are normalized by the ratio of the turbine diameter to the product of the free stream wind speed and lateral integration length, $1/U_\infty^3 L$, where $L = D$ in this case. The vertical dashed line denotes the edges of the central rotor volumes.

the magnitudes between the two surfaces can be attributed to the tower's presence and other measurement noise. Hence, the
discussion will focus on $S_6$ as the representative case. The positive pitch case shows the highest magnitudes in the first volume.
The injection of momentum steadily decreases as the dominant CVP convects towards the windward direction. Compared to
the baseline case, the flux steeply decreases leading up to the second rotor due to the upwind loaded induction. The baseline
pitch case transitions from a positive flux as the wake expands to a negative value due to the asymmetric wake profile shown in
Section 4.3, where flow is injected on the leeward side. This negative flux grows minimally in maximum magnitude between
the volumes, from 0.02 to 0.03 in Volumes 1 and 3. Conversely, the negative pitch case shows a close to zero flux across
the surface across all volumes. Although the wake is ejected out axially above the rotor, as shown in Section 4.3, the strong
asymmetry in the wake and dominance of lateral influx of momentum yields minimal vertical velocity magnitudes on the upper
and lower surfaces of the rotor.

Overall, the fluxes of the Reynolds stress terms, illustrated in Figure 19, are an order of magnitude lower than that of the
advective, demonstrating the dominance of the advective terms over the Reynolds stresses in the recovery of kinetic energy in
the farm. For all pitch cases, the highest magnitudes are concentrated near the wakes of the rotor. For the positive pitch, the
largest magnitude flux is on $S_3$ due to the ejection of the wake. Unlike the case with the advective terms, the baseline case



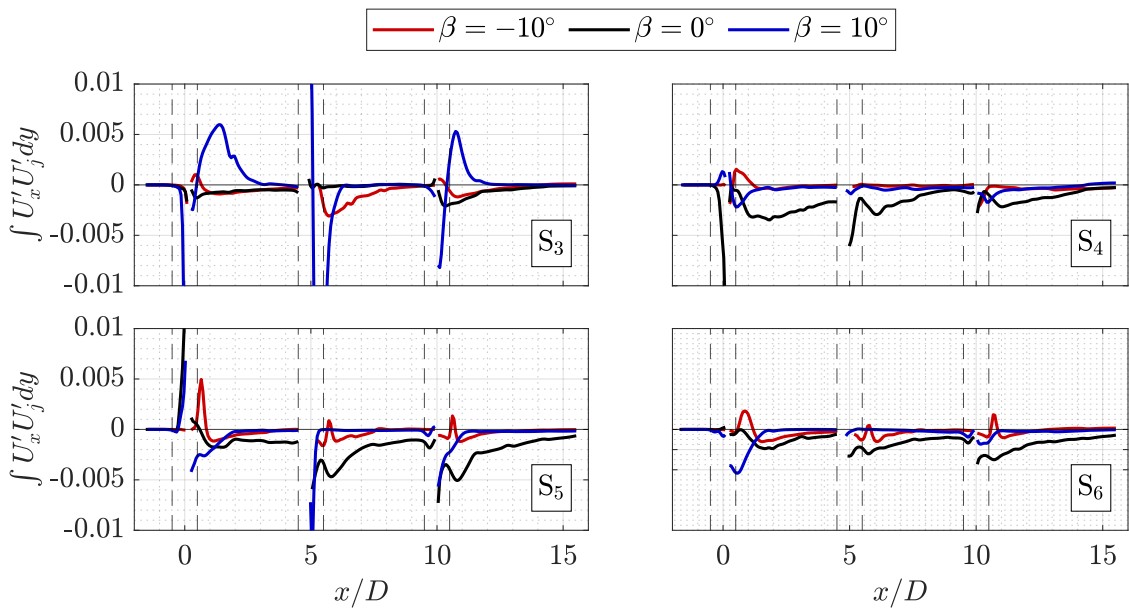

**Figure 19.** Line integrated Reynolds stress flux terms as a function of downstream position $x/D$ through the surfaces $j = S_{3,4,5,6}$, as noted in Figure 17 for the three modes of operation. The flux terms are integrated within the lateral boundaries $-0.5 \leq y/D \leq 0.5$. All values fluxes are normalized by the ratio of the turbine diameter to the product of the free stream wind speed and lateral integration length, $1/U_\infty^3 L$, where $L = D$ in this case. The vertical dashed line denotes the edges of the central rotor volumes.

exhibits higher magnitudes in flux on the upper and lower surfaces of the control volume $(S_5, S_6)$ compared to the pitched modes of operation. This highlights the relative importance of turbulent mixing towards the wake recovery for the baseline
operation of the rotor.

     The discussion above gives an indication of the surface fluxes as a function of the downstream location. The surface-integrated fluxes for each volume are shown in Figure 20. Here, the sum of the advective and Reynold stress contributions towards the flux is considered, as dictated in Equation (7). In addition, the total flux of each surface over all three volumes is shown in the bottom tile of Figure 20.

As discussed in relation to Figure 18, the integrated fluxes of $S_5$ and $S_6$ remain similar due to symmetry for the three pitch cases. The negative pitch case shows a discrepancy in Volumes 2 and 3 with opposite signs of minimal magnitude. For the positive pitch case, the flux through $S_3$ decreases between Volume 1 and 3 due to the presence of the wake of the neighboring column. The vertical flux through $S_6$ also decreases in magnitude from 0.05 to 0.03 between Volumes 1 and 3. For the negative pitch case, the flux magnitude injection from $S_4$ increases from 0.04 to 0.06 between Volumes 1 and 3. In general, the largest
fluxes for the baseline case are evident on the axial control surfaces, growing from 0.01 to 0.03 between Volumes 1 and 3, reaching a similar magnitude to the positive pitch case.



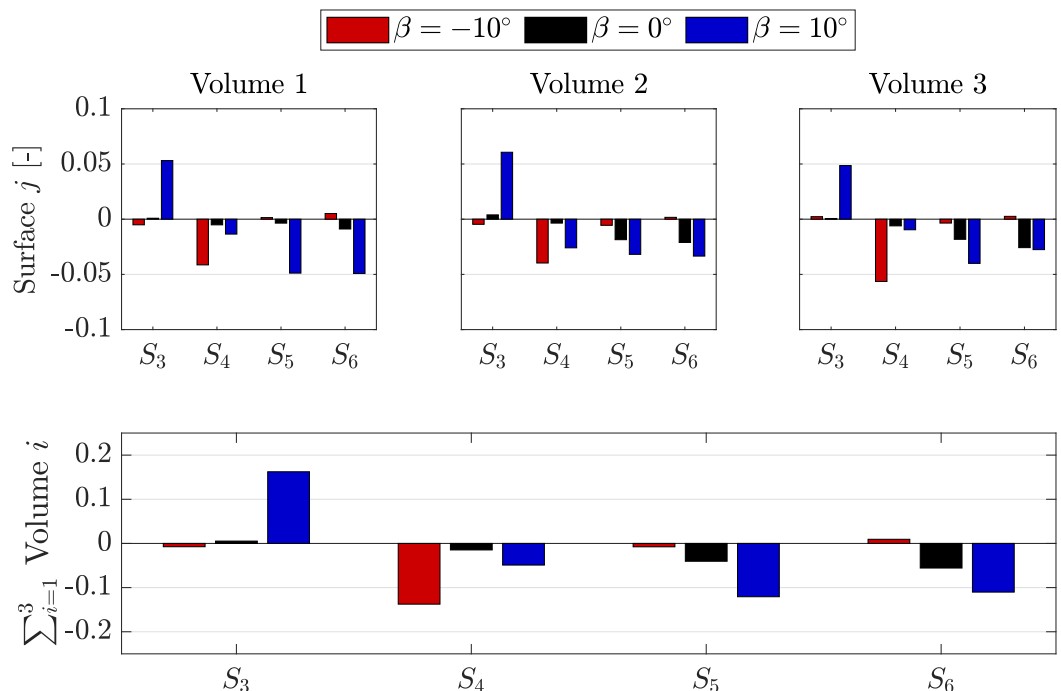

**Figure 20.** Surface integrals of the control surfaces ($j = 3, 4, 5$) as described in Equation (7), where Volume 1, 2, and 3 ($i$) correspond to the volumes behind the first, second, and third rotors, respectively. The sum of surface fluxes over all three volumes is shown in the bottom plot. All values are normalized by the $1/U_\infty^3 S_i$. The surface areas between all four surfaces are constant, with $S_i = 4D^2$.

Considering the sums between all volumes for each control surface, it is evident that the positive pitch case exhibits the highest magnitude in flux, both with the ejection of the wake on $S_3$ and injection of high momentum flow from $S_6$. The negative pitch case demonstrates the highest momentum injection through the leeward surface $S_3$. The baseline case demonstrates

minimal flux across the lateral surfaces, with higher magnitudes in the vertical injection of momentum from $S_6$, attributed to the domain CVP discussed in Section 4.2.

To give a visual representation of the fluxes in the wind farm volume, the sum of the axial and lateral advective contributions towards the flux at discrete cross-stream planes $x/D = 2.5, 7.5, 12.5$ for the three modes of operation is shown in Figure 21. The vectors show the in-plane velocity components, and the projected frontal area of the central rotor column and wind farm

grid size is shown via solid and dashed lines, respectively. In this case, the magnitude of the in-plane velocity components is taken to compute the advective flux.

Consistent with Figure 20, the positive pitch case demonstrates higher magnitudes in advective flux at all cross-stream locations. However, the largest magnitudes are present at the symmetry plane of the rotor $z/D = 0$, where the wake is ejected out laterally, driven by the dominant lateral flow component as indicated by the vectors. This trend remains consistent on





the windward side of the rotor. Meanwhile, on the leeward side, the wake of the neighboring column connects within the measurement domain and projected frontal area of the rotor at $x/D = 12.5$, as evidenced in Figure 10. However, as this wake has low magnitudes in streamwise flow, the advective transport of momentum is lower.

For the negative pitch case, larger magnitudes in advective momentum entrainment are present on the leeward side of the rotor, which grows in magnitude between the volume behind the first and second rotor in the column. This is consistent with

the vortex topology discussed in Section 4.2.

Finally, for the baseline case, the advective flux is low behind the first rotor due to the lack of a dominant vortex pair in the wake, as shown in Section 4.2. Behind the second and third rotors, advective flux is concentrated above and below the projected rotor surface, and the dominant cylinder-like vortex pair entrains momentum. However, the magnitude is not as high as the positive pitch case as the streamwise flow components are lower in magnitude due to the wake incident by the first rotor

in the column, as shown in Figure 10.

To shed light on the total advective momentum flux within the farm grid area, the average sum of lateral and axial advective flux as a function of height $z/D$ and cross-stream location $x/D$ within the lateral domain $-1.6 \pm y/D \pm 1.6$ is shown in Figure 22. Consistent with the results above, the positive pitch case demonstrates the highest magnitudes in advective flux, growing in size when progressing further downstream in the farm. High magnitudes of flux are concentrated around the top

and bottom of the rotors as high momentum flow is injected into the volume. Although the magnitudes at mid-span are lower, they are still notable as the wake is ejected out latterly. The negative pitch case demonstrates minimal mean advective flux behind the first rotor but grows in magnitude to approximately 1.2 behind the second and third. The shape confirms that most of the advection of flux occurs at the mid-span of the rotor. As expected, the baseline pitch case exhibits minimal magnitudes as flux, with no significant wake deflection. Marginally larger fluxes are apparent at the upper and lower surfaces of the rotor

due to the dominant cylinder-like CVP commented on in Section 4.2.

## 5 Conclusions

This study provides the first experimental database of 3D resolved time-averaged flowfield measurements for a wind tunnel scale high-energy density wind farm made of VAWTs with and without wake control. The simulated farm consisted of nine rotors arranged in a grid fashion with fixed lateral and streamwise spacing of 3.18D and 5D, respectively. In addition to the

wake measurements for a baseline case, the wakes for two cases using the "vortex generator" wake control strategy, namely with blade pitches of $\beta = -10°$ and $\beta = 10°$, were captured to highlight the re-generative wind farming capability of a VAWT farm.

The theory behind the "vortex generator" VAWT wake control strategy using fixed-blade pitch was described, highlighting the expected modes of re-energization in the wake for cases where the blades are pitched in versus pitched out. The modes

of operation of the wake control strategy were confirmed by the streamwise vortex topology within the farm. An energization of the upwind and downwind vortices is achieved by pitching the blade inward and outwards, respectively. For the former case, a significant influx of momentum from above and below the rotor was observed in the wake topology in parallel with a



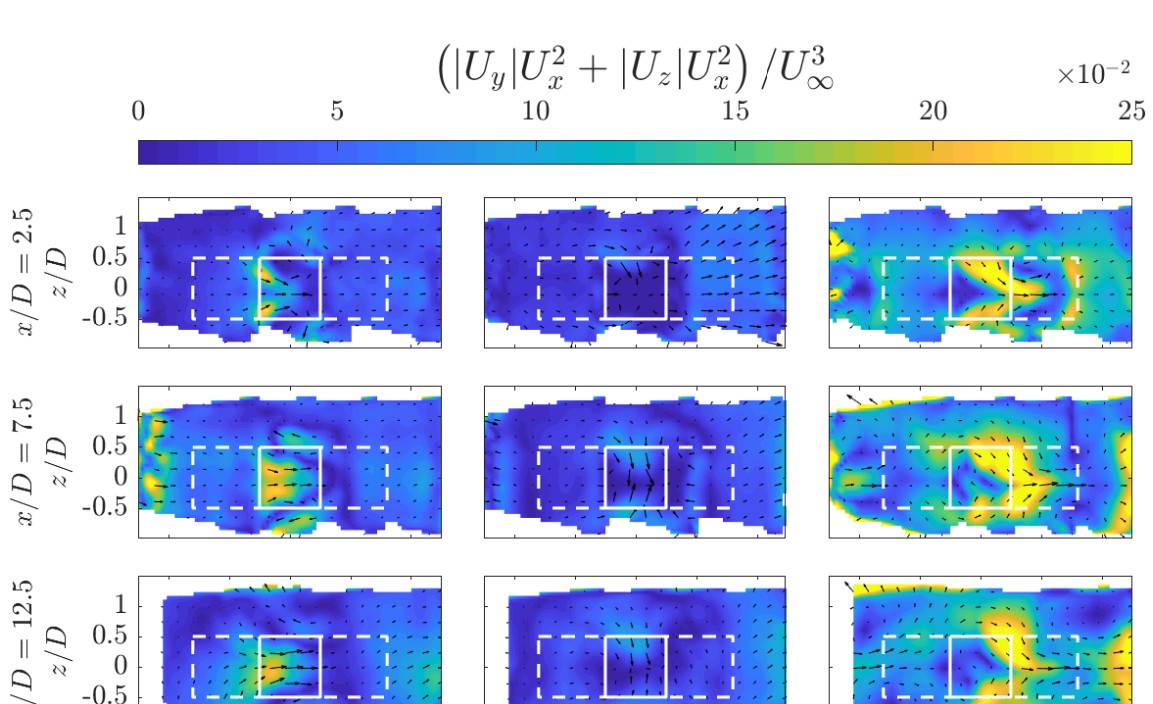

**Figure 21.** The sum of the lateral and axial advective contributions towards the flux at discrete cross-stream planes $x/D$ = 2.5, 7.5, and 12.5 for the three pitch cases. The magnitudes of the velocity components $|U_y|$ and $|U_z|$ are considered to isolate the regions of momentum transfer. All values are normalized by the free-stream velocity component $U_\infty^3$. In-plane velocity vectors are also shown. The box delineated by the solid white line denotes the projected frontal area of the central rotor column, and the white dashed line the grid size of the farm $-1.6 \leq y/D \leq 1.6$.

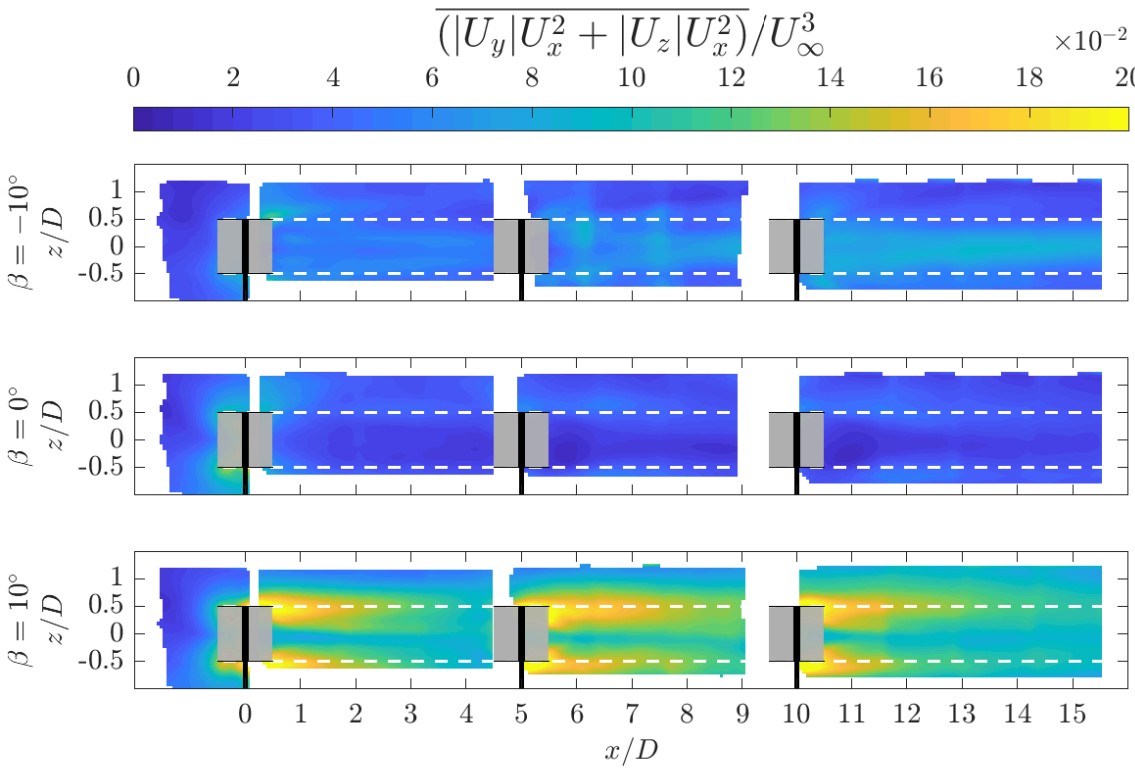

**Figure 22.** Average sum of the lateral and axial advective contributions towards the flux as a function of height $z/D$ and cross-stream location $x/D$. The fluxes are averaged over the lateral domain $-1.6 \leq y/D \leq 1.6$ and normalized by the free stream flow $U_\infty^3$. The gray-shaded region shows the rotors of the central column.

significant lateral deflection of the wake, consistent with thrust measurements of an isolated rotor. For the latter case where the blades are pitched out, the downwind energized vortices induce an upwash of the wake while injecting high momentum flow

from the sides of the rotor.

The efficacy of these modes towards wake re-energization was highlighted by considering a proxy of the available power in the wake. The authors once again re-iterate that as the power of the rotors was not measured, the balance between the penalty associated with the steering turbine and that gained in the farm as a whole was not evaluated. The exaggerated pitch angles were selected to stay consistent with prior work performed by Huang (2023) and to magnify the benefits associated

with modifying the rotor loading with wake recovery in mind. However, such large pitch angles are likely not to be applied for full-scale rotors, especially positive pitches, given the unsteady load fluctuations stemming from the large degree of flow separation in the upwind passage (Le Fouest and Mulleners (2024)). Nonetheless, these results demonstrate a proof-of-concept



of the vortex generator mode for VAWT wake re-energization of a farm scale. When considering the case of a potential rotor directly in line with the central column, an available power recovery for the positive pitch case reaches a maximum of 72.4% at $x/D = 3.2$, which is a factor of 6.4 higher than the baseline case. Meanwhile, the recovery within the farm for the negative pitch case reaches a maximum of 53% at $x/D = 8.8$ (approx. 3D downstream of the second rotor), with a factor of 2.1 higher than the baseline case. This analysis was extended to allow for lateral offsets of hypothetical downstream rotors in the farm, highlighting the high potential for staggered arrangements. Although the positive pitch case showed the highest recovery rate for the inline rotor case, given the lateral deflection of the wake, the regions of high AP diminish on the windward side of the central column. These lucrative regions are further diminished on the farm when the wakes from the neighboring columns begin to fall within the projected area of the rotor column. Finally, the re-energization mechanism of the farm was highlighted by adapting a volume analysis of the mean kinetic energy equation. The fluxes through the lateral and axial surfaces of the central rotor column confirm the working principle of the vortex generator mode and shed light on its efficacy on a farm scale.

The methodology and analysis presented can be extended in several directions, including the investigation of alternative turbine layouts (staggered vs. in-line), the inclusion of a boundary layer, ground effect, and turbulent inflow. Furthermore, the database is valuable for the development and validation of numerical models in the context of "wind farming". Future efforts should be applied to quantifying the trade-offs between power loss in the steering turbine and the gain in the farm as a whole at a variety of operating conditions for VAWTs to select more realistic passive and active pitch offsets for wake recovery.

## Appendix A: Vorticity fields

As described in Section 2, the "vortex generator" mode of operation for VAWTs for wake control relies mainly on the modification of the streamwise vorticity $\omega_x$, which has been thoroughly discussed in Section 4.2. Nonetheless, for completeness, the other vorticity components, namely the axial $\omega_y$ and lateral $\omega_z$, as shown in Figure A1 and Figure A2, respectively.

The lateral vorticity components $\omega_y$ for the baseline case $\beta = 0°$ show a strong symmetry between the upper and lower halves of the rotor with clockwise and anti-clockwise rotating structures, respectively. Such behavior is also observed by phase-locked measurements of Tescione et al. (2014) and field experiments of Wei et al. (2021). For the first rotor, the vortex structure transitions from an inboard movement (towards $z/D = 0$) on the leeward half towards an outboard movement on the windward half. A similar trend has been observed by Tescione et al. (2014). The subsequent rotors show a similar trend with overall shorted vortex lengths due to the reduction in loads of the rotors. The vortex structure from the leeward side seems to extend further than those on the windward, presumably a product of the reduction in load on the windward side due to the asymmetric wake of the first rotor, as discussed in Section 4.3.

For the positive pitch case $\beta = 10°$, the directions of the dominant vortices remain the same as the baseline case. The structures deflect towards the windward side, consistent with the observations and discussion in Section 4.2. * What are these other structures at see at just at the tips. Something from the struts, perhaps. Same for the negative pitch case as well.

The axial vorticity component $\omega_z$ is driven by the vortices shed along the span of the blades. For the baseline case, the dominant structure is clockwise and anti-clockwise on the windward and leeward sides, respectively. This is consistent with



**Figure A1.** Iso-surfaces of the lateral vorticity contours at $\omega_y D/U_\infty = \pm 1$ for pitch cases $\beta = \pm 10°$, and $\beta = 0°$. The rotors are shown at a $\theta = 0°$ phase, with the gray-shaded cylinders representing the idealized actuator cylinder load.



phase-locked measurements obtained by Tescione et al. (2014). As in the aforementioned case, the lengths of these structures are reduced for the subsequent rotors due to the reduction in load. On the windward side, the positive shear layer is also present, concentrated around the upper section of the rotor. This can be attributed to the asymmetric load and wake profile shown in Section 4.3, where a switch between the dominant contributor towards curl in the z-direction $dU_x/dy$ would switch twice near

the upper windward edge of the wake (given the mushroom-like shape). This would be asymmetric on the bottom half of the rotor but is absent due to the limited data on the bottom half of the rotor.

For the positive pitch case, the structures deflect laterally. Given the increase in the severity of the mushroom-like shape compared to the baseline case, the second positive shear layer on the windward side is also present and remains a feature for the subsequent rotors as well with the asymmetric pair. Finally, for the negative pitch case, the shear layer on the windward

side is deflected outwards whilst that on the leeward side is contracted inward towards the core of the rotor as flow is injected more heavily for the subsequent rotors in the farm.

*Data availability.* The data that support the findings of this study will be made openly available in 4TU ResearchData.

*Author contributions.* DB developed the methodology, carried out the experiment, performed the analysis, and wrote the manuscript. JM contributed towards the development of the methodology, and carried out the experiment. AS revised the manuscript and provided scientific

supervision during the experiment, analysis, and documentation phase. CF provided guidance towards the methodology, experiment, and analysis phase.

*Financial support.* This research has been supported by the funding received from the European Union's Horizon 2020 research and innovation program under grant agreement No 101007135 as part of the project - XROTOR.

*Competing interests.* The authors report no conflict of interest.

*Acknowledgements.* We wish to thank the support of Ed Roessen and Rob van der List (Dienst Elektronische en Mechanische Ontwikkeling - AE Faculty) for the design and manufacturing of the rotors.Furthermore, thank you to Gert-Jan Berends (Aerodynamics Laboratory technician) for his critical role in the conceptualization and execution of the wind farm electronic systems. Finally, a special thank you to Delphine De Tavernier for her comprehensive review of the manuscript.



**Figure A2.** Iso-surfaces of the axial vorticity contours at $\omega_z D/U_\infty = \pm 1$ for pitch cases $\beta = \pm 10°$, and $\beta = 0°$. The rotors are shown at a $\theta = 0°$ phase, with the gray-shaded cylinders representing the idealized actuator cylinder load.



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
