# Peer review of "Experimental demonstration of regenerative wind farming using a high-density layout of VAWTs"

_Wind Energy Science, 2024_

## Referee Comment (RC2)

*Review on: Experimental demonstration of regenerative wind farming using a high-density layout of VAWTs by Bensason et al.*

**General comments**

The subject addressed by the authors is of great interest in the context of wind farm development. It is also original in that it deals with the potential of vertical-axis wind turbines, which are little studied in the literature.

The paper is very well written and structured. The analytical approach is very good and well explained. Particularly noteworthy is the careful presentation of the experiment, which is of very high quality. The measurement of wake velocity on such a large scale is uncommon in the literature, and the data obtained are used in a very relevant way.

The paper seems to me to be, after the answers to a few questions raised below, quite publishable.

**General questions**

- Would the advantages of the global pitch control proposed here be the same if the turbines were placed in a turbulent atmospheric boundary layer? Would the diffusion of generated vortices result in a loss of control efficiency?

- The disadvantage of reproducing wind turbines on a small scale is the value of the Reynolds number based on the chord. Do the authors have any idea of the influence of Reynolds number on the observed wake dynamics?

- With regard to working scales, is it possible to identify similarities between the size of the models used and the results obtained (wake deflection, turbulent entrainment, etc.)? If we want to use this study to optimize the placement of wind turbines in a farm situation, a scaling by a characteristic length scale of the wind turbine could be interesting.

**Detailed questions**

- Line 225: The blockage is given. But it seems to me that the wind turbines are positioned at the edges of the wind tunnel jet. Does this affect the results obtained? In particular on the rows of wind turbines on the sides?

- Line 259: A spatial resolution of 19 mm is given for velocity measurements. This represents approximately 8 to 10% of the wind turbine diameter. So we can have 15 points of velocity value on one diameter. Is this sufficient for the purposes of the study?

- Line 393: It is written that the structures are 'shorter'. Isn't that very precise? Does it refer to a size or an intensity, a circulation?

- Line 390-393: How do the structures generated by the wind turbines evolve spatially? Is there any interaction between them? Does the relative position of the turbines has to be important?

---

## Author Response (AR1)

**Manuscript ID: WES-2024-177**

Experimental demonstration of regenerative wind farming using a high-density layout of VAWTs

March 2025

The authors thank the reviewers for their time and valuable insight. Their feedback has been helpful in improving the quality of this manuscript. The following document contains the point-by-point rebuttal to the comments of Reviewer 1 and Reviewer 2. The reviewers' comments are listed below in BLACK, with subsequent responses to each in RED.

**Reviewer 1**

This manuscript aims to experimentally investigate the three-dimensional, time-averaged wake flow of a VAWT farm, both with and without wake control. By utilizing volumetric particle tracking velocimetry, the study clearly identifies how the mean velocity field of the VAWT wake varies with the pitch of the blades. The effectiveness of the vortex modes is highlighted by analyzing the available power coefficient ($f_{\mathrm{AP}}$), even though the power output of the rotors was not measured. The wake flow topology is thoroughly described and compared for three different blade pitch angles (0° and ±10°). It is important to note that the non-zero pitch values used in this study are relatively large and may not fully replicate in-situ conditions.

While this analysis is quite interesting, especially for examining the physical aspects of wake flow re-energization, as well as for mean wake modeling, several technical questions are raised throughout the article.

1. I agree that the wake produced by turbines during energy extraction affects the performance of downstream turbines, making the arrangement of turbines a crucial factor in influencing flow behavior. However, to highlight the impact of both vortex generator modes, the authors have chosen to consider an in-line turbine arrangement. Don't the authors believe that this arrangement is particularly advantageous for analyzing the wake effects related to the present blade pitch?

Previous analyses (Shen et al. 2024. Ocean Engineering, 311, 118965; Azadani 2023. Ocean Engineering, 272, 113855 and references therein) did not observe that in-line turbine arrangement is the best arrangement for optimized farm performance. Please discuss on the present turbine arrangement choice.

Would vortex generator modes be as efficient for turbine staggered arrangement? Please comment.

Thank you for these insightful comments. Indeed, previous studies have investigated the performance of VAWT farms with in-line and staggered arrangements. Simulated results by Hezaveh et al. (2018b) reported increases on the order of 20-25% of average wind farm $C_p$ with turbine spacings ranging from 5D to 20D for aligned and staggered arrangements. Furthermore, an in-depth analysis by Hezaveh and Bou-Zeid (2018a) shed light on the mean kinetic energy replenishment mechanisms of different VAWT farm arrangements, once again confirming the performance increase throughout the farm with staggered arrangements.
Several arrangements could be tested in the wind tunnel using the experimental procedure described in this study. However, the objective here was to highlight the potential of the passive blade pitch recovery on a less favorable farm arrangement (in-line and high density) where losses would be high for the baseline case. Whether the in-line arrangement of staggered would be best for cases where the vortex generator modes are used was elucidated when discussing Figure 15 and Figure 16. The results confirm the benefit of a staggered

arrangement for the baseline case $\beta = 0°$ with a close to symmetric availability on both the windward and leeward sides. Meanwhile, the negative pitch case $\beta = -10°$ shows a gradual deflection of the wake towards the windward side, suggesting a favorable staggered arrangement with rotors towards the leeward side of that upwind. The positive pitch case $\beta = 10°$ demonstrates the largest degree of wake deflection towards the windward side, again suggesting a staggered arrangement would be favorable.

The justification for choosing an in-line arrangement has been added to Section 1 (Introduction) in the last paragraph.

2. The Reynolds number is quite small compared to in-situ conditions. Is there any Reynolds number effect on the turbulent VAWT wake flow? Also, mean flow representations look like laminar-type flow. Please comment.

Thank you for this observation. Indeed, a common limitation in lab-scale wind turbine wake studies (of both HAWTs and VAWTs) is the satisfaction of the Reynolds number similarity. The wake of a full-scale (outdoor) H-type VAWT was measured by Wei et al. (2021) at a diameter-based Reynolds number of $1.2 \times 10^6 \leq Re_D \leq 1.8 \times 10^6$. The wake shape was compared with the results of previous water-channel measurements performed at $Re_D = 8 \times 10^4$, confirming a high degree of similarity. These results confirm the observations of Parker and Leftwich (2016) who measured the wake behind a lab-scaled VAWT in a wind tunnel at Reynolds numbers ranging from $Re_D = 6 \times 10^4$ to $Re_D = 1.8 \times 10^5$. The authors report that whilst the maximum flow deficit behind the rotor increases with the Reynolds number (by 17% between the extremes), the overall wake structure is unchanged.

With regard to the impact of Reynolds number on the turbulence, the work of Bachant and Wosnik (2016) measured the wake behind an H-type VAWT at diameter based on Reynolds number ranging between $4 \times 10^5 \leq Re_D \leq 12 \times 10^5$. The results concluded that there was a minimal impact on the overall wake structure. An analysis of the kinetic energy transport quantities in the axial and lateral directions (similar to that done in Section 4.6) concluded marginally lower levels of turbulent transport with increased $Re_D$, with no clear impacts on the advective transport. Though the reasons for this were not explored, it is likely due to blade-scale unsteady flow effects as laminar separation bubbles and the formation and shedding of dynamic stall vortices in the wake, which are more severe at lower Reynolds numbers, impacting the turbulence in the wake and overall performance consistent with the measured the lower $C_P$ at lower Reynolds number reported.

Finally, the numerical simulations by Boudreau and Dumas (2017) were performed at $Re_D = 10^7$. An analysis of the streamwise wake recovery budget was performed using the time-averaged URANS equation and concluded a dominance of the advective transport compared to the turbulence by order of magnitude, whilst viscous effects (diffusion and dissipation) are negligible, consistent with the reported results in Section 4.6 of the present study.

Hence, although the Reynolds number used in this study of $Re_D = \rho U_\infty D / \mu = 6.1 \times 10^4$ is lower than in-situ conditions, it is within a similar range as those tested in previous wind tunnel studies which have concluded Reynolds number independence on the wake shape. A note about the working Reynolds number here compared to previous studies has been added to Section 3.4 of the revised manuscript.

Regarding the laminar-type flow of the mean flow representations, it is important to note that all the wakes shown are spatially and time-averaged over a 9.2s acquisition period, equating to approximately 102 turbine revolutions, as described in Section 3.3.

3. 3D Measurement method and post-processing.

The mean flow analysis relies mainly on the efficiency of the flow measurement method and its associated post-processing method.

(a) Line 257-258: 'Given an overlap of 75%, the final grid spacing of the velocity vectors is 19mm' The overlapping mesh grid resolution appears to be quite high compared to the flow scales being considered, particularly near the rotor, which has a chord length of 30 mm. What is the validity of the resulting time-averaged mean velocity field in the near wake of the vertical axis wind turbine (VAWT)?

Thank you for this observation. Indeed, the resulting vector resolution of the data processing is 19.9mm, corresponding to approximately 15 vectors distributed across the rotor diameter. The objective of the study is not to resolve the cyclic trailing tip vortex formation and convection in the near wake of a VAWT. Such a study was reported by Caridi et al. (2016), who had a chord to a grid resolution of 7.5, as opposed to the current study with 1.6. Rather, the objective of this study is to evaluate the macro-scale wake dynamics of the farm (on the wake scale), of which this resolution is sufficient. Time-averaged results reported by Huang et al. (2023a) for an identical isolated rotor and pitch offset as in this study reported trailing vortex sizes in the wake on the order of 5 to 10 chord lengths at $x/D = 3$. This suggests that the chosen resolution of this study is sufficient to capture the large-scale vortices and dynamics of the scaled wind farm. A convergence study between the selected sub-volume size (in voxels) on the wake structure is shown in an added section, Appendix A, concluding an acceptable quantification of the wake shape with the given vector spacing.

(b) What about the spatial smoothing effect over the voxel on the mean results?

Thank you for this question. In this work, we use a second-order polynomial regression to the velocity distributions in each sub-volume to resolve the mean velocity gradients. This technique was proposed by Agüera et al. (2016), who provide a comprehensive overview of the spatial smoothing within a sub-volume on the mean and turbulent statistics of PTV data. The authors report that the polynomial fit technique is approximately equivalent to working with a 3 times smaller interrogation window size.

(c) Globally, what is the size of the smallest resolved turbulent flow scales?

Thank you for this question. In relation to the previous comment, the smallest resolved turbulent flow scale will be on the order of 3 times smaller than the integration volume. As noted in Section 3.3, this is 80mm. Hence, the smallest resolved turbulent flow scale is on the order of 27mm.

(d) What are the limitations of present velocity measurements?

Thank you for this comment. In addition to the previous questions regarding the treatment of turbulent flow scales and the overall wake topology specific to this study, and flow measurement uncertainty described in Section 3.5, a comprehensive overview of the limitations of 3D Lagrangian Particle Tracking can be found in the work of Schröder and Schanz (2023).

(e) Could the authors detail the computation of the 'mean' wx and wz vorticity? The resulting vorticity field seems to approach a laminar-type flow field.

As clarified in the previous comments, the data considered in the results are averaged in time, and in space within the interrogation bin, hence the smooth flow structures shown in Figure 7 and Figure 9. The mean vorticity is computed by taking the curl of the mean flow components using a three-point central finite difference scheme. This has been clarified in the revised manuscript in Section 4.2.

(f) The authors present a volume analysis of the energy equation for which the fluxes of the Reynolds stresses Ux'Uz' and Ux'Uy' are determined. Could the authors discuss the determination of the Reynolds tensor components? What is the effect of the spatial flow structure smoothing in the voxel on the fluctuating velocity results?

Thank you for this comment. The Reynolds stresses $R_{ij}$ are computed using Equation (1), taken from the DaVis 10.2 Software Product Manual, as

$$R_{ij} = \overline{V_i'V_j'} = \frac{1}{N-1}\sum_{i=1}^{N}(V_{i,n} - \overline{V_i})(V_{j,n} - \overline{V_j}), \tag{1}$$

where $i, j = x, y, z$, $N$ is the number of available vectors, overline values denote average velocity components, and those without instantaneous values.

The impact of the spatial smoothing is elucidated in the revised manuscript in Appendix A by looking at the turbulent kinetic energy $k$. Here, a range of sub-volume sizes with vector spacings ranging from 8.1mm to 23.6mm is presented at the symmetry plane $z/D = 0$ in the wake of the first rotor at $x/D = 3.5$ for the baseline case $\beta = 0°$. The discussion concludes that the largest difference to a more resolved wake is in the shear layer, whilst minor differences are observed within the projected frontal area of the rotor. When considering the root mean square of the difference between the chosen sub-volume size of 128 voxels and the most resolved case of 52 voxels, the regions in the shear layer show a relative error on the order of 10%. Once again, details on the impact of the second-order polynomial smoothing within the voxel (bin) on the turbulence statistics are discussed by Agüera et al. (2016).

(g) Could the authors provide an estimation of the error of quantities plotted in Figures 19 and 20?

Thank you for this comment. First, a correction has been made to Section 3.5. Initially, an average of N = 900,000 samples was used in Equation 1, which is the number of tracks considered in a bin once spatially and temporally averaged. However, this averages to a significant number of tracks per bin per time instant, which are not necessarily uncorrelated. Hence, $N = 4600$ (number of recordings) is assumed as the number of uncorrelated samples per bin, which is a conservative estimate. Hence, Equation 1 yields a maximum of $\epsilon_x = 0.02\,\mathrm{m\,s^{-1}}$, which is 0.67% of the freestream velocity $U_\infty = 3\,\mathrm{m\,s^{-1}}$.

As discussed in Section 4.6, the advective contributions towards the flux of streamwise kinetic energy are close to two orders of magnitude higher than the turbulent, shown in Figures 18 and 19, respectively. Hence, the uncertainties related to the advective fluxes would be the most critical when evaluating the overall surface fluxes shown in Figure 20. The uncertainty of the fluxes $\epsilon_{,\mathrm{flux}}$ shown in Figure 18 are expressed as follows:

$$\epsilon_{\mathrm{flux}} = \sqrt{(U_j U_x \epsilon_x)^2 + (0.5 U_x^2 \epsilon_j)^2} \tag{2}$$

where the subscript $j$ is either $y$ for surfaces 3 and 4 and $z$ for surfaces 5 and 6, and $\epsilon_j$ is the velocity uncertainty, as described in Equation 1 in Section 3.5. Hence, the uncertainty in the line-integrated flux $U_{\mathrm{I,Line}}$ is expressed as:

$$U_{\mathrm{I,Line}} = \sqrt{\sum (\epsilon_{\mathrm{flux}} \Delta j)^2} \tag{3}$$

where $\Delta j$ is the grid size in the $y$ and $z$ direction (which is uniform as 19.9mm). Representative values of normalized (by $U_\infty^3 L$, where $L = D$) uncertainty in the flux range between $0.6 \times 10^{-3}$ and $0.8 \times 10^{-5}$. When considering a representative value normalized flux in Figure 18 of 0.08, the relative error is on the order of 0.75% to 1%.

Finally, considering the uncertainties associated with the advective contributions for the surface integrals of the flux in Figure 20 (top three tiles), the uncertainty $U_{\mathrm{I,Surface}}$ is computed as:

$$U_{\mathrm{I,Surface}} = \sqrt{\sum (\epsilon_{\mathrm{flux}} \Delta j^2)^2} \tag{4}$$

and the next across all volumes $U_{\mathrm{I,Volume}}$ shown in Figure 20 (bottom tile) is:

$$U_{\mathrm{I,Volume}} = \sqrt{(U_{\mathrm{I,Surface,V1}})^2 + (U_{\mathrm{I,Surface,V2}})^2 + (U_{\mathrm{I,Surface,V3}})^2} \tag{5}$$

where the subscripts V1, V2, and V3 correspond to the surface fluxes of the measurement volumes, which are the first, second, and third turbine rows, respectively. Here, the relative uncertainties are found to be in a similar range as above. This order of magnitudes in error has been added to the revised manuscript in the caption of Figure 18.

(h) Can the present velocity measurements account for turbulence? If so, a key factor that affects rotor performance and blade fatigue is the turbulent kinetic energy. Could the author display and analyze this quantity in the wake flow of the VAWT as well as in front of the rotors?

Thank you for this comment. Yes, the present velocity measurements can account for turbulence by taking the fluctuating components (denoted with primes) with respect to the spatially and temporally averaged velocities in a given bin. These are the components used in the discussion related to Figure 19, which computes the line-integrated Reynolds stresses along the control surfaces of the wind farm as a function of the wake position. As discussed in Section 4.6, the relative contributions towards the flux streamwise kinetic energy due to turbulence are an order of magnitude lower compared to the advective process. This is especially true for the pitched cases, which rely heavily on the induced cross-flows generated by the energized streamwise vortices. Nonetheless, cross-stream planes of the turbulent kinetic energy $k$ at the same planes shown in Figure 8 and Figure 10 are added and discussed in the Appendix, named Appendix C, in the revised manuscript.

These issues need to be addressed to ensure the reliability of the results.

4. Figures 14-15: While the turbine induction process is clearly indicated in front of the first turbine, it seems not to be visible for the other turbine apart from the second turbine with beta=-10°. Is it a consequence of the measurement method and post-processing tools? Please comment.

Indeed, the induction process is clear for the first rotors when considering Figure 15, namely the reduction and increase in induction for the negative and positive pith cases, respectively. This can be attributed to the re-distribution of the rotor load towards the downwind and upwind halves of the cycle for the two cases, as discussed in Section 2. This is highlighted further when considering the available power upstream of the first rotor in Figure 14, which is increased for the negative pitch case. The impacts of the induction of the second and third rotors in the column are also visible when considering Figure 14, namely by the decreases in available power upstream of the rotors, rather than a gradual increase as seen behind the final row as the wake fully returns to the neutral conditions.

5. Mean velocity analysis.

Numerous analyses of mean flow were conducted to examine the recovery of the wake and the available power. Instead of simply plotting spatial average quantities, it would be more effective to provide additional representations of the mean streamwise velocity over the rotor area, 2D upstream of the rotors. This approach would allow us to assess not only the overall impact of inflow shear on the development of the rotor wake but also its possible influence on blade structural fatigue. Inflow shear is expected to cause asymmetric loading, which could consequently affect the fatigue of the blades. Consequently, alongside the analysis of the mean available power, discussing the predictability of turbine lifespan in relation to inflow shear would be valuable.

Thank you for this comment. Indeed, changes in inflow angle would impact not only the blade loading but also the development of the wake and structural fatigue of the rotor. The "vortex generator" strategy induces significant cross-flows in the wake, leading to large lateral and axial flow components, as discussed in Sections 4.2 - 4.4. The mean streamwise velocities at measurement planes, 2.5D upstream of the second and third rotor and 2.5D downstream of the third rotor are shown in Figure 10. Furthermore, vectors are included to shed light on the in-velocities, which quantify the shear and yaw of the flow without the farm. These vectors are included for the same planes in Figure 8. Finally, Figures 11 and 12 show the cross-flow vectors at the symmetry planes $y/D = 0$ and $z/D = 0$, respectively. These provide insight into the skewed and yawed flow within the farm. Linking the available power to the predicted turbine lifespan falls outside the scope of this study. However, as commented on in Section 5, future research efforts should be aimed at quantifying the trade-offs between power performance and wake steering gains to optimize passive and active pitching schemes. Including structural considerations such as unsteady blade load and torque, fluctuations are also valuable, along with the considerations of rotor loading on a potential downstream rotor. This has been added to the discussion in Section 5.

**Reviewer 2**

**General Comments**

The subject addressed by the authors is of great interest in the context of wind farm development. It is also original in that it deals with the potential of vertical-axis wind turbines, which are little studied in the literature.

The paper is very well written and structured. The analytical approach is very good and well explained. Particularly noteworthy is the careful presentation of the experiment, which is of very high quality. The measurement of wake velocity on such a large scale is uncommon in the literature, and the data obtained are used in a very relevant way.

The paper seems to me to be, after the answers to a few questions raised below, quite publishable.

**General Questions**

1. Would the advantages of the global pitch control proposed here be the same if the turbines were placed in a turbulent atmospheric boundary layer? Would the diffusion of generated vortices result in a loss of control efficiency?

   Thank you for the comment. As with HAWTs, the inclusion of turbulence would reduce the efficacy of the steering strategy as the flow wake would intrinsically recover faster, and the dominant counter-rotating vortex pair would diffuse faster. Given the limitations of the wind tunnel facility, a scaled atmospheric boundary layer and turbulence could not be tested. However, numerical simulations (RANS) have been carried out by Huang (2023b) for an isolated VAWT, and an array of two and three inline rotors using passive pitch adjustments of the same magnitude and streamwise spacing of 4D with a turbulence intensity of $I = 8\%$. The results conclude significant overall power gains of the farm on the order of 37% - 45% when all rotors are pitched by $\beta = -10°$ or $\beta = 10°$.

   The inclusion of a boundary layer for a VAWT farm using the "vortex generator" strategy has not been studied to date. However, the work of Rolin and Porté-Agel (2018) sheds light on the impact of a ground and scaled boundary layer on the development of an H-type VAWT wake. In the near-wake $(x/D = 1)$, the dominance of the upwind-generated streamwise vortices is clear and symmetric to a high degree about the $z/D = 0$ plane. As the wake develops, the vortices generated by the bottom tip diffuse and deform to such a degree that they are no longer discernible, likely due to the induced stretching of the upper vortex from the UW quadrant. As a result, the UW vortex remains the only dominant structure from $x/D = 6$ onward, inducing a significant lateral outwash of the wake on the windward side. This asymmetry along the plane $z/D = 0$ would persist for pitched cases as well. The study of the wake development for an isolated rotor using the "vortex generator" under different conditions, such as an atmospheric boundary layer, inflow turbulence, and ground, is an area of future research. A discussion of these possible research directions has been noted in Section 5 (Conclusions).

2. The disadvantage of reproducing wind turbines on a small scale is the value of the Reynolds number based on the chord. Do the authors have any idea of the influence of Reynolds number on the observed wake dynamics?

   Thank you for the comment. A similar comment was given by Reviewer 1 and is addressed in the answer to Reviewer's comment 2.

3. With regard to working scales, is it possible to identify similarities between the size of the models used and the results obtained (wake deflection, turbulent entrainment, etc.)? If we want to use this study to optimize the placement of wind turbines in a farm situation, a scaling by a characteristic length scale of the wind turbine could be interesting.

   Thank you for this comment. In this study, the working length scale assumed is the diameter of the wind turbine $D$, which is used consistently for the normalization of the spatial components of the plots

and is the common method in literature Hezaveh and Bou-Zeid (2018a). A comprehensive review of VAWT wake scaling is provided by Huang et al. (2022), who shed light on a universal scale length for VAWTs with aspect ratios different from 1.

**Detailed Questions**

1. Line 225: The blockage is given. But it seems to me that the wind turbines are positioned at the edges of the wind tunnel jet. Does this affect the results obtained? In particular on the rows of wind turbines on the sides?

   Thank you for the comment. Indeed, the turbines at the edge of the grid are positioned 33cm from the edge of the wind tunnel (from blade tip to edge). The lateral spacing of $y/D = 3.18$ was used to ensure sufficient distance from the edge of the turbine to the edge of the jet whilst keeping a reasonable spacing. With this setup, there is a $1.1D$ spacing between the edge of the rotor and the edge of the stream tube (assuming no expansion beyond the projected frontal area of the tunnel outlet). The study of Lignarolo et al. (2014) showed that the turbulence intensity remains below approximately 3% at this position. However, at approximately 0.1m ($0.3D$) from the edge, the flow becomes more turbulent, with a turbulence intensity of 10%. When considering the results of Huang et al. (2023a), who measured the wake of the same rotor (isolated) at the same pitch magnitudes, the wake deflection in the lateral direction is minimal for the $\beta = 0°$ and $\beta = -10°$ cases, falling within $-0.50 \leq y/D \leq 0.75$ at $x/D = 5$. However, it is more pronounced for the $\beta = 10°$ cases, ejecting to approximately $y/D = 1.2$ on the windward side. Hence, we can expect that the wakes of the rotors in the left most column for the positive pitch case would interact with the shear layer of the open jet and perhaps steer the stream tube beyond $x/D = 5$. Despite this, the focus of this experiment was to investigate the wake topology of the central column in a farm setting. Future research efforts are underway to numerically validate the experimental case (3x3 grid) and an infinite farm case using the "vortex generator" recovery mode. This will shed light on the impact of the neighboring columns, tunnel blockage, and relevance toward an infinite farm wake topology. This limitation has been addressed in Section 3.4 and Section 5 (Conclusions) in the revised manuscript.

2. Line 259: A spatial resolution of 19 mm is given for velocity measurements. This represents approximately 8 to 10% of the wind turbine diameter. So we can have 15 points of velocity value on one diameter. Is this sufficient for the purposes of the study?

   Thank you for this comment. A similar remark has been made by Reviewer 1 (see comment 3a) and addressed by providing a grid convergence study in Appendix A of the revised manuscript.

3. Line 393: It is written that the structures are 'shorter'. Isn't that very precise? Does it refer to a size or an intensity, a circulation?

   Thank you for this comment. Indeed, this term is not precise. This is in reference to the Iso-surface plot of the trailing vorticity (Figure 7), specifically for the positive pitch case $\beta = 10°$. The iso-surfaces are flooded values of $\omega_x D/U_\infty = \pm 0.4$, and hence only appear when a certain intensity is surpassed. Hence, a shorter structure would not indicate an absence of trailing vorticity further in the wake but rather a higher degree of diffusion (reduced intensity). This has been clarified in the text of the revised manuscript.

4. Line 390-393: How do the structures generated by the wind turbines evolve spatially? Is there any interaction between them? Does the relative position of the turbines has to be important?

   Thank you for this comment. The spatial evolution of the streamwise vortices generated by the first, second, and third rotors are visualized in Figure 7. Here, we can observe the interactions between those generated by the different rotors in the column, as well as with the neighboring column (specifically for the positive pitch case). The cross-stream planes presented in Figure 8 include labels of the dominant structures, linking them back to the generation quadrant described in Section 2. This plot sheds further light on the convection of the vortical structures vertically and laterally.

**References**

Agüera, N., Cafiero, G., Astarita, T., and Discetti, S.: Ensemble 3D PTV for high resolution turbulent statistics, Measurement Science and Technology, 27, 124 011, 2016.

Bachant, P. and Wosnik, M.: Effects of Reynolds number on the energy conversion and near-wake dynamics of a high solidity vertical-axis cross-flow turbine, Energies, 9, 73, 2016.

Boudreau, M. and Dumas, G.: Comparison of the wake recovery of the axial-flow and cross-flow turbine concepts, Journal of Wind Engineering and Industrial Aerodynamics, 165, 137–152, 2017.

Caridi, G. C. A., Ragni, D., Sciacchitano, A., and Scarano, F.: HFSB-seeding for large-scale tomographic PIV in wind tunnels, Experiments in Fluids, 57, 1–13, 2016.

Hezaveh, S. H. and Bou-Zeid, E.: Mean kinetic energy replenishment mechanisms in vertical-axis wind turbine farms, Physical Review Fluids, 3, 094 606, 2018a.

Hezaveh, S. H., Bou-Zeid, E., Dabiri, J., Kinzel, M., Cortina, G., and Martinelli, L.: Increasing the power production of vertical-axis wind-turbine farms using synergistic clustering, Boundary-layer meteorology, 169, 275–296, 2018b.

Huang, M.: Wake and wind farm aerodynamics of vertical axis wind turbines, Ph.D. thesis, ISBN 9789055841745, https://doi.org/10.4233/uuid:14619578-e44f-45bb-a213-a9d179a54264, 2023b.

Huang, M., Ferreira, C., Sciacchitano, A., and Scarano, F.: Wake scaling of actuator discs in different aspect ratios, Renewable Energy, 183, 866–876, 2022.

Huang, M., Sciacchitano, A., and Ferreira, C.: On the wake deflection of vertical axis wind turbines by pitched blades, Wind Energy, 2023a.

Lignarolo, L., Ragni, D., Krishnaswami, C., Chen, Q., Ferreira, C. S., and Van Bussel, G.: Experimental analysis of the wake of a horizontal-axis wind-turbine model, Renewable Energy, 70, 31–46, 2014.

Parker, C. M. and Leftwich, M. C.: The effect of tip speed ratio on a vertical axis wind turbine at high Reynolds numbers, Experiments in Fluids, 57, 1–11, 2016.

Rolin, V. F. and Porté-Agel, F.: Experimental investigation of vertical-axis wind-turbine wakes in boundary layer flow, Renewable energy, 118, 1–13, 2018.

Schröder, A. and Schanz, D.: 3D Lagrangian particle tracking in fluid mechanics, Annual Review of Fluid Mechanics, 55, 511–540, 2023.

Wei, N. J., Brownstein, I. D., Cardona, J. L., Howland, M. F., and Dabiri, J. O.: Near-wake structure of full-scale vertical-axis wind turbines, Journal of Fluid Mechanics, 914, A17, 2021.

---

## Referee Report (RR1)

The reviewer would like to thank the authors for the important work they have done in response to the comments and questions posed by the two reviewers.

The authors answered the various questions and comments with precision and honesty.

The authors have considered the reviewers' remarks about the need for further studies to validate the effectiveness of their device in the presence of an atmospheric boundary layer, or in the case of turbine arrangements other than 'in line'. Adding a perspective on these points in the conclusion helps to avoid too rapid a generalization of the results and gains achieved.

Regarding the influence of Reynolds number, in relation to the scale at which the tests are carried out, the references added seem convincing to me. Nevertheless, I think there may be an effect of Reynolds number on wake development, particularly as a function of blade profile. Some profiles being more sensitive to Reynolds number.

For these reasons, the new version of the manuscript seems to me entirely suitable for publication in Wind Energy Science.